# Aquificae overcomes competition by archaeal thermophiles, and crowding by bacterial mesophiles, to dominate the boiling vent-water of a Trans-Himalayan sulfur-borax spring

**Nibendu Mondal**[1¤], **Subhajit Dutta**[1], **Sumit Chatterjee**[1], **Jagannath Sarkar**[1], **Mahamadul Mondal**[1], **Chayan Roy**[2], **Ranadhir Chakraborty**[3], **Wriddhiman Ghosh**[1]*

**1** Department of Biological Sciences, Bose Institute, Kolkata, India, **2** Department of Plant and Environmental Sciences, University of Copenhagen, Copenhagen, Denmark, **3** Department of Biotechnology, University of North Bengal, Siliguri, India

¤ Current address: International Institute of Innovation and Technology, Kolkata, India
* wriman@jcbose.ac.in, Wriman@rediffmail.com

**Data Availability Statement:** The shotgun metagenome sequence dataset was submitted to the Sequence Read Archive (SRA) of the National

## Abstract

Trans-Himalayan hot spring waters rich in boron, chlorine, sodium and sulfur (but poor in calcium and silicon) are known based on PCR-amplified 16S rRNA gene sequence data to harbor high diversities of infiltrating bacterial mesophiles. Yet, little is known about the community structure and functions, primary productivity, mutual interactions, and thermal adaptations of the microorganisms present in the steaming waters discharged by these geochemically peculiar spring systems. We revealed these aspects of a bacteria-dominated microbiome (microbial cell density ~8.5 × $10^4$ mL$^{-1}$; live:dead cell ratio 1.7) thriving in the boiling (85°C) fluid vented by a sulfur-borax spring called Lotus Pond, situated at 4436 m above the mean sea-level, in the Puga valley of eastern Ladakh, on the Changthang plateau. Assembly, annotation, and population-binning of >15-GB metagenomic sequence illuminated the numeral predominance of Aquificae. While members of this phylum accounted for 80% of all 16S rRNA-encoding reads within the metagenomic dataset, 14% of such reads were attributed to Proteobacteria. Post assembly, only 25% of all protein-coding genes identified were attributable to Aquificae, whereas 41% was ascribed to Proteobacteria. Annotation of metagenomic reads encoding 16S rRNAs, and/or PCR-amplified 16S rRNA genes, identified 163 bacterial genera, out of which 66 had been detected in past investigations of Lotus Pond's vent-water via 16S amplicon sequencing. Among these 66, *Fervidobacterium*, *Halomonas*, *Hydrogenobacter*, *Paracoccus*, *Sulfurihydrogenibium*, *Tepidimonas*, *Thermus* and *Thiofaba* (or their close phylogenomic relatives) were presently detected as metagenome-assembled genomes (MAGs). Remarkably, the *Hydrogenobacter* related MAG alone accounted for ~56% of the entire metagenome, even though only 15 out of the 66 genera consistently present in Lotus Pond's vent-water have strains growing in the laboratory at >45°C, reflecting the continued existence of the mesophiles in the ecosystem. Furthermore, the metagenome was replete with genes crucial for thermal adaptation in the

Center for Biotechnology Information (NCBI), USA, under the BioProject PRJNA296849, with the BioSample and Run accession numbers SAMN31749247 and SRR22364517 respectively. The bacterial and archaeal 16S amplicon sequence datasets were submitted to the SRA, under the BioProject PRJNA296849, with BioSample accession numbers SAMN32724325 and SAMN32724260, and Run accession numbers SRR23072972 and SRR23072970, respectively.

**Funding:** This research was financed by Bose Institute (via intramural faculty grants) as well as the Science and Engineering Research Board (SERB), Government of India (GoI) (SERB grant number EMR/2016/002703). N.M. received fellowships from SERB and Bose Institute. S.D. and J.S. obtained their fellowships from Council of Scientific and Industrial Research, GoI. S.C. and M. M. received a fellowship from the Department of Biotechnology, GoI. The funders played no role in the study design, data collection and analysis, decision to publish, or preparation of the manuscript.

context of Lotus Pond's geochemistry and topography. In terms of sequence similarity, a majority of those genes were attributable to phylogenetic relatives of mesophilic bacteria, while functionally they rendered functions such as encoding heat shock proteins, molecular chaperones, and chaperonin complexes; proteins controlling/modulating/inhibiting DNA gyrase; universal stress proteins; methionine sulfoxide reductases; fatty acid desaturases; different toxin-antitoxin systems; enzymes protecting against oxidative damage; proteins conferring flagellar structure/function, chemotaxis, cell adhesion/aggregation, biofilm formation, and quorum sensing. The Lotus Pond Aquificae not only dominated the microbiome numerically but also acted potentially as the main primary producers of the ecosystem, with chemolithotrophic sulfur oxidation (Sox) being the fundamental bioenergetic mechanism, and reductive tricarboxylic acid (rTCA) cycle the predominant carbon fixation pathway. The Lotus Pond metagenome contained several genes directly or indirectly related to virulence functions, biosynthesis of secondary metabolites including antibiotics, antibiotic resistance, and multi-drug efflux pumping. A large proportion of these genes being attributable to Aquificae, and Proteobacteria (very few were ascribed to Archaea), it could be worth exploring in the future whether antibiosis helped the Aquificae overcome niche overlap with other thermophiles (especially those belonging to Archaea), besides exacerbating the bioenergetic costs of thermal endurance for the mesophilic intruders of the ecosystem.

## Introduction

Hydrothermal environments, despite being the potential nurseries of early life on Earth [1–3], are characterized by the entropic disordering of biomacromolecular systems, which in turn restricts the habitability of these ecosystems [4,5]. Thermophilic and hyperthermophilic microorganisms [6], that are adept to counter chaos and restore order at the molecular level without incurring untenable energy costs for cell system maintenance, are the habitual natives of hydrothermal vents. For most other taxa abundant at the 30–40°C sites of these ecosystems (these are the phylogenetic relatives of mesophilic microorganisms), habitability is progressively constrained in the 50°C to 100°C trajectory [5,7–10]. Conversely, as the bioenergetic cost of cell system maintenance get relaxed down the temperature gradient, habitability, and therefore *in situ* diversity, increases for most microbial taxa except for the true thermophiles [11,12].

Idiosyncratic to the low-biodiversity community structure axiomatic for hydrothermal vent ecosystems, a geochemically unusual category of Trans-Himalayan hot springs, located on the either side of the tectonically active Indus Tsangpo Suture Zone (ITSZ, the collision intersection between the Asian and Indian continental crusts engaged in Himalayan orogeny) in eastern Ladakh, India [13–16], has recently been reported for their extra-ordinarily diversified vent-water microflora [5,10,17,18]. Compared with other well studied hydrothermal systems, the geochemical specialty of these Trans-Himalayan hot springs lies in their paucity of dissolved solids in general and calcium and silicon in particular, which again is accompanied by the exceptional abundance of boron, chlorine, sodium, and various sulfur species including both sulfide and sulfate. Concurrent to such reports of highly bio-diverse microbiomes from the hot springs of the Puga valley and Chumathang geothermal areas situated on the Changthang plateau, to the south and north of the ITSZ respectively, studies of microbial diversity within the vent-waters, and vent-adjacent areas, of a number of hot springs located across

discrete Himalayan and Trans-Himalayan geothermal regions have corroborated that hydrothermal environments are not "Thermophiles only" territories [19–26].

As for the well-studied Puga hydrothermal system, PCR-amplified 16S rRNA gene sequence-based investigations of microbial communities within the boiling vent-waters of a number of springs have revealed the presence of several such genera whose members are incapable of laboratory-growth at >45°C [10,17,18]. Diurnal fluctuations in the biodiversity and chemistry of the vent-waters, in conjunction with circumstantial geological evidences, have indicated that the copious tectonic faults of this ITSZ-adjoining area serve as potential conduits for the infiltration of meteoric water, and therewith mesophilic microorganisms, to the shallow geothermal reservoir [10]. The phenomenon of mesophilic microorganisms getting introduced to hot spring systems has been reported from other parts of the world too; for instance, systemic infiltration of mesophiles from surface and sub-surface sources of cold water and meteoric fluids have been observed for hot springs of New Zealand and Yellowstone National Park, USA [27,28]. Furthermore, for the Puga hot spring system, shotgun metagenome sequencing-based investigation of microbial mat communities along the spring-water transits and dissipation channels of a microbialite has identified several genus-level populations to occupy such sites within the hydrothermal gradients which have temperatures tens of degree-Celsius above the *in vitro* growth-limits of those taxa [5,29]. While the overall geomicrobiology of the Puga hot springs have indicated the system's peculiar chemistry (mineralogy) to be a key facilitator of the high habitability of the environment [5,18], physiological attributes of a vent-water isolate identified dissolved solutes such as lithium and boron, which are characteristic to the Puga hydrothermal system [15,16], as the key enablers of high temperature endurance by the mesophilic bacteria present *in situ* [30].

Despite a considerable progress in our basic awareness about the peculiar geobiology of Trans-Himalayan sulfur-borax hot springs, we still do not have any quantitative idea about the structural and functional dynamics of the vent-water microbiomes of these astrobiologically significant hydrothermal systems [16,31,32]. For instance, we have a fair measure of the microbial diversity in terms of the variety of taxa present, but we are completely uninformed about the relative abundance and temporal consistency of the individual taxa constituting the microbiome, the thermal adaptations of the native microorganisms (especially the phylogenetic relatives of mesophilic bacteria present), and the mutual interactions of the various components of the microbiome. We also know nothing about the pathways of primary productivity that sustain the ecosystem in these boiling spring waters. To address these shortcomings in our knowledge on the ecology of Trans-Himalayan sulfur–borax spring systems, this study comprehensively reveals the vent-water microbiome of the Lotus Pond hot spring, which is one of the most conspicuous hydrothermal manifestations within the Puga valley [10,16,17]. While the bacterial diversity of Lotus Pond's vent-water has been investigated previously by means of 16S amplicon sequencing [10,17], the present exploration, for the first time, unraveled the qualitative as well as quantitative architecture of the microbiome via shotgun metagenome sequencing, and fluorescence microscopy. Besides elucidating the dominance and equitability of taxa within the microbiome, the >15 giga base metagenomic sequence data generated was analyzed to illuminate the aforesaid aspects of microbiome functioning. Furthermore, to know whether the Lotus Pond vent-water microbiome remained consistent over time, its bacterial and archaeal diversity was evaluated afresh by amplified 16S rRNA gene sequencing and comparing the current findings with the previous data obtained on the basis of similar analyses.

## Materials and methods

### Study site and sampling

Lotus Pond, located at 33˚13'46.3"N and 78˚21'19.7"E, is one of the most prominent hot springs or geysers within the Puga valley (Fig 1A), which in turn is one of the most vigorous geothermal areas of the Indian Trans-Himalayas. Notably, Puga Valley is situated at an altitude of 4436 m, where the boiling point of water remains at around 85˚C [10,16,33]. The venting point (Fig 1B) is embedded in a hot water pool that has been formed by its own discharges and lies within the old crater (Fig 1A) which was formed from the same hydrothermal eruption that gave rise to the geyser itself. From one side of the hot water pool, the spring-water flows over thin unstable terraces of geothermal sinters, into the Rulang rivulet, which meanders through the valley of Puga. On the other flank of the hot water pool, a wall of travertine, sulfur, and boron mineral deposits [10,16] guard the crater and debars the spring-water from escaping the pool (Fig 1C).

Water (having temperature 85˚C and pH 7.5) was collected as described at length previously [5,10,17,18] from Lotus Pond's vent at 13:00 h of 29 October 2021, when ebullition was strong and fumarolic activity copious. For microscopic studies, 500 mL vent-water was passed through a sterile 0.22 μm mixed cellulose ester membrane filter (Merck Life Science Private

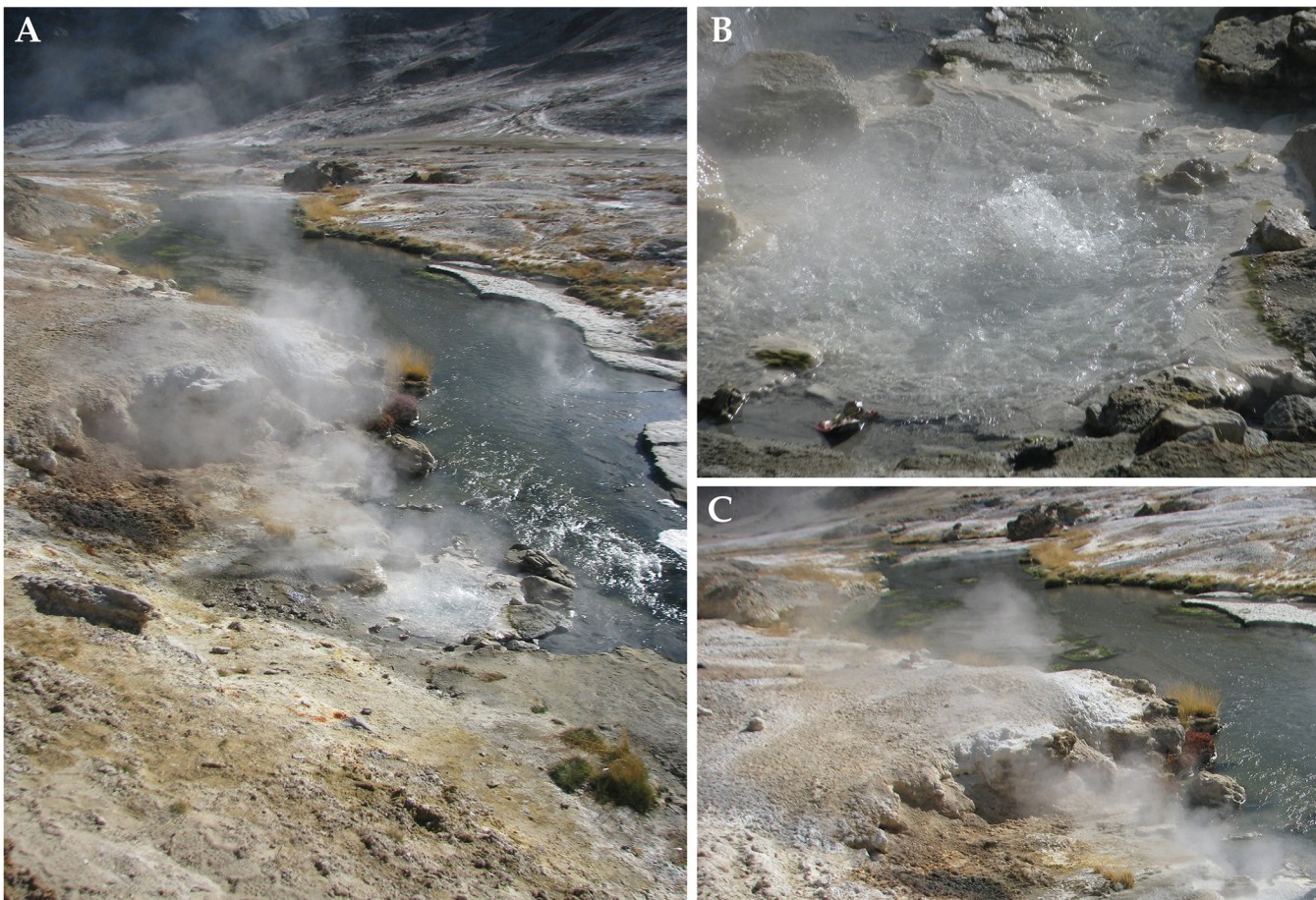

**Fig 1.** Topography of the study site: (**A**) the Lotus Pond hot spring in the context of the river Rulang and the valley of Puga, (**B**) the boiling vent-water explored for microbiome structure and function, (**C**) boratic sinters, and fresh condensates of sulfur and boron minerals covering the broken wall of the old crater within which Lotus Pond is embedded.

Limited, India) having a diameter of 47 mm, and inserted into a sterilized cryovial containing 5 mL of autoclaved 0.9% (w/v) NaCl solution supplemented with 15% (v/v) glycerol. Simultaneously, for the extraction of total environmental DNA (metagenome), overall 5 L vent-water was passed through five individual sterile 0.22 μm filters, each of which was inserted into a separate sterilized cryovial containing 5 mL of autoclaved 50 mM Tris:EDTA (pH 7.8). After filter insertion, all the cryovials were sealed with parafilm (Tarsons Products Limited, India), packed in polyethylene bags, put into dry ice, and air lifted to the laboratory at Bose Institute in Kolkata. The entire on-field procedure was carried out aseptically using sterilized syringes (Tarsons Products Limited, India), Swinnex filter holders (Merck KGaA, Germany), and forceps on every occasion of water collection from the vent, filtration through the membrane, and insertion of the filter to the cryovial.

## Enumeration of microbial cells

Total microbial cell count per unit volume of vent-water was determined alongside the number of metabolically active and metabolically inactive cells present. The microbial cells that were filtered out from 500 mL vent-water were first detached from the adherence matrix by shredding the filter with sterilized scissors, and then vortexing for 30 min, within the NaCl-glycerol containing cryovial in which the filter was inserted on-field. The vial was spun at 1000 $g$ for 5 seconds to allow the filter shreds to settle down. Finally, the supernatant was collected (without disturbing the bottom debris), its exact volume measured, and then analyzed as described previously [34] using a hemocytometer (Paul Marienfeld GmbH & Co. KG, Germany) and an upright fluorescence microscope (Olympus BX53 Digital, Olympus Corporation, Japan). Microbial cell density measurement was carried out based on the axiom that the final supernatant collected contained all the microbial cells that were present in the 500 mL vent-water filtered on-field. The cell suspension (i.e. the collected supernatant) was stained with 4′6-diamidino-2-phenylindole (DAPI), fluorescein diacetate (FDA) or propidium iodide (PI) solution, and cell numbers were counted as described previously [35]. Details of the procedure followed for microscopy and microbial cell enumeration have been given in the Supplementary Methods.

## Metagenome extraction and sequencing

The 0.22 μm membrane filters suspended in sterile 50 mM Tris:EDTA (pH 7.8) were treated as described previously [5,10,17,18], and from the final cell suspension obtained metagenomic DNA was extracted using PureLink Genomic DNA Mini Kit (Thermo Fisher Scientific, USA) following the manufacturer's protocol. DNA molecules having high levels of structural integrity, as evident from NanoDrop spectrophotometry and Qubit fluorometry (both the operations were carried out using platforms manufactured by Thermo Fisher Scientific), were used as the input material (approximately 1 ng total quantity) for preparing shotgun metagenome sequencing library with the help of Nextera XT DNA Library Prep Kit (Illumina Inc., USA). Quality of the final library was assesed using high sensitivity D1000 screen tape in 2200 TapeStation (Agilent Technologies Inc., USA), while its quantification was carried out using a Qubit fluorometer. Paired end (2 × 250 nucleotides) sequencing of the library was performed in a Novaseq 6000 (Illumina Inc.) next-generation DNA sequencer. The whole metagenomic sequence dataset of 31353866 read-pairs obtained in the process was submitted to the Sequence Read Archive (SRA) of the National Center for Biotechnology Information (NCBI), USA, under the BioProject PRJNA296849.

In tandem with the aforesaid procedure, potential traces of contaminating microial DNA, often referred to as the "kitome" [36,37], was isolated in the same way as described above for

the five membrane filters through which vent-water was passed on-field. The only difference in case of the kitome preparation was that no vent-water was passed through any of the five filters, which in this case also were inserted to five individual sterilized cryovials each containing 5 mL of autoclaved 50 mM Tris:EDTA (pH 7.8). The final suspension obtained for the blank filters was subjected to the same DNA extraction procedure using the same PureLink Genomic DNA Mini Kit as it was done for the actual sample suspension. No DNA could be detected using NanoDrop spectrophotometry or Qubit fluorometry, so the so-called kitome DNA preparation could not be subjected to shotgun sequencing on the Illumina platform.

## Assembly and annotation of the metagenomic sequence

Clipping of adapters and quality-filtering of the trimmed sequences (for average Phred score ≥20) were carried out using Trim Galore v0.6.4 (https://github.com/FelixKrueger/TrimGalore) with default parameters. The 31342225 processed read-pairs obtained post quality filtering were subjected to *de novo* assembly by Megahit v1.2.9 using default parameters [38]. Open reading frames (ORFs) or genes were identified within the assembled contigs using Prodigal v2.6.3 with default parameters [39]. Annotation of the predicted ORFs into protein coding sequences (CDSs) was carried out by searching against the eggNOG database v5.0 [40] using eggNOG-mapper v2.1.9 in default mode using Diamond algorithm [41]. The software eggNOG-mapper aligns CDSs to precomputed orthology assignments in the eggNOG database in such a way that it can infer orthologs, and transfer the functional annotations and taxonomic information obtained to the queries, thereby providing detailed taxonomic annotations for the individual metagenomic CDSs analyzed [41]. Notably, CDS annotated in this way included potential pseudogenes, which could also be outcomes of sequencing error-induced frame shifts. The eggNOG-derived data was processed to assign orthology identities, and determine pathway affiliations, for the annotated CDSs with the help of the information available in the literature in tandem with those provided in the Kyoto Encyclopedia of Genes and Genomes (KEGG; https://www.genome.jp/kegg/). Furthermore, to identify the clusters of orthologous genes (COGs) to which the different ORFs predicted within the assembled metagenome belonged, the translated form of the Prodigal-derived gene catalog was annotated by searching against the COG Little Endian version of the Conserved Domain Database of NCBI located at https://ftp.ncbi.nih.gov/pub/mmdb/cdd/little_endian/Cog_LE.tar.gz, using COG-classifier v1.0.5 [42]. The bacterial version v7.0 of the antibiotics & Secondary Metabolite Analysis Shell (antiSMASH) pipeline was used in strict detection mode [43,44] to identify gene clusters for secondary metabolites biosynthesis within the assembled metagenome of Lotus Pond. To identify antibiotic resistance genes, the assembled metagenome was searched against the Comprehensive Antibiotic Resistance Database (CARD) v3.2.8 using the software Resistance Gene Identifier v6.0.3 with default parameters [45]. Here, gatifloxacin, levofloxacin, moxifloxacin, nalidixic acid, norfloxacin, sparfloxacin, and ciprofloxacin antibiotics were grouped as fluoroquinolones; cloxacillin and oxacillin were grouped under beta-Lactam antibiotics.

## Direct annotation of metagenomic reads corresponding to rRNA gene sequences

The 62684450 quality-filtered reads available for the Lotus Pond metagenome were searched against the rrnDB database v5.8 [46] using Bowtie2 v2.2.5 [47] in "—end-to-end" mode to extract sequences matching prokaryotic 16S rRNA genes. The aligned FASTQ file was converted to FASTA format and used for sequence annotation with the help of RDP Classifier located at http://https://rrndb.umms.med.umich.edu/estimate/. While 82756 aligned reads

were extracted from Lotus Pond's metagenomic sequence dataset via mapping against rrnDB, their eventual annotation using RDP Classifier with confidence cut-off 0.8 led to the taxonomic classification of 81619 reads.

## Construction and taxonomic characterization of metagenome-assembled genomes (MAGs)

The *de novo* assembled metagenomic contigs that were ≥2500 nucleotides in length were first binned into putative population genomes or MAGs, via three separate procedures using three different softwares with default parameters: Metabat2 v2.12.1 [48], MaxBin2 v2.2.4 [49], and CONCOCT v1.1.0 [50]. Best quality MAGs were then selected using DASTool v1.1.6 [51] with default parameters from the outputs of the three different binning procedures. The MAGs selected were checked for their completeness and contamination level with the help of CheckM v1.2.2 by searching against collocated sets of marker genes that are ubiquitous and single-copy within a given phylogenetic lineage [52].

Taxonomic identity of the selected MAGs was first predicted independently via searches for genom-genome relatedness conducted using Genome Taxonomy Database Toolkit (GTDB-Tk) v1.7.0 [53], Rapid Annotations using Subsystems Technology (RAST) [54], and Type Strain Genome Server (TYGS) in default mode [55]. As and when required, MAG versus genome/MAG pairs were tested for the extent of their *in silico* or digital DNA-DNA hybridization (dDDH) using the online software Genome-to-Genome Distance Calculator (GGDC) v3.0 [55]. In TYGS as well as GGDC analyses, total length of all high-scoring segment pairs (HSPs) divided by the total genome length (TGL) of the query MAG was considered as the dDDH value. Taxonomic classifications inferred using the aforesaid index, i.e. "(total HSP length) / (TGL)", were further corroborated by testing for the index "(total identities found in HSPs) / (total HSP length)". Notably, these two indices are also known as TYGS formula $d_0$ / GGDC formula 1, and TYGS formula $d_4$ / GGDC formula 2, respectively. Orthologous genes-based average nucleotide identity (ANI) between a pair of MAG and genome/MAG was calculated using ChunLab's online ANI Calculator [56].

The MAG sequences were deposited to the GenBank and annotated for their gene contents using the Prokaryotic Genome Annotation Pipeline (PGAP; https://www.ncbi.nlm.nih.gov/genome/annotation_prok/) of the NCBI, USA. Phylogeny of bacterial MAGs, in relation to their closest genomic relatives, was reconstructed based on conserved marker gene sequences, using the Up-to-date Bacterial Core Gene Pipeline v3.0 [57] with default parameter. The dataset used for this purpose contained all the close genomic relatives of the individual MAGs that were indicated by GTDB-Tk and TYGS analyses. The best fit nucleotide substitution model was selected using ModelTest-NG v0.1.7 [58], following which a maximum likelihood tree was generated using RAxML v8.2.12 [59] with 10,000 bootstrap experiments. The software utility called Interactive Tree of Life v6.7.5 [60] was used to visualize the phylogram graphically. The KEGG automatic annotation server (KAAS) was used for ortholog assignment and pathway mapping for the CDSs identified within the MAGs [61]. Relative abundance of the individual MAGs within the metagenome was determined by mapping all quality-filtered metagenomic read-pairs available onto the individual MAG sequences via independent *in silico* experiments carried out using Bowtie2 v2.2.5 [47] in "—end-to-end" mode.

## PCR-amplified 16S rRNA gene sequence analysis

PCR-amplified 16S rRNA gene sequence-based taxonomic diversity was revealed using the fusion primer protocol [10,17,22] on the Ion S5 DNA sequencing platform (Thermo Fisher Scientific), which in turn involved PCR amplification, sequencing, and analysis of the V3

regions of all Bacteria-specific 16S rRNA genes and V4-V5 regions of all Archaea-specific 16S rRNA genes [18] present in the Lotus Pond metagenome. The forward primer 5′ -CCTAC GGGAGGCAGCAG- 3′ and the reverse primer 5' -ATTACCGCGGCTGCTGG- 3′ were used to amplify the bacterial V3 regions, while the archaeal V4-V5 regions were amplified using the forward and reverse primers 5′ -GCYTAAAGSRICCGTAGC- 3′ and 5′ -TTTCAGYCTTGC GRCCGTAC- 3′ respectively [18]. Albeit no quantifiable DNA was present in the so-called kitome preparation, it was still subjected to individual PCRs using the two 16S rRNA gene sequence primer-pairs mentioned above. Notably, no DNA could be detected even after re-PCR with any of the two primer-pairs used, so the null PCR products were not subjected to sequencing on the Ion S5 platform.

Reads present in an amplicon sequence dataset (725025 and 589009 reads were there in the bacterial and archaeal sets respectively) were trimmed for their adapters, filtered for average Phred score ≥20 and length >100 nucleotides, and then clustered into operational taxonomic units (OTUs), i.e. species-level entities united by ≥97% 16S rRNA gene sequence similarities, by USEARCH v10.0.259 [62]. While singleton sequences were eliminated before clustering, OTUs generated for a given dataset were classified taxonomically based on their consensus sequences using the RDP Classifier with confidence cut-off 0.8. Rarefaction curves (S1 Fig in S1 File) were generated via USEARCH v10.0.259.

## Results

### Density of microbial cells in Lotus Pond

Gross density of microbial cells in the vent-water of Lotus Pond was found to be approximately $8.5 \times 10^4$ mL$^{-1}$, via staining with DAPI, followed by hemocytometry using an upright fluorescence microscope (Fig 2). Using similar instrumentation, the density of metabolically active cells was estimated to be approximately $5.4 \times 10^4$ mL$^{-1}$, via staining with FDA, while that of apparently dead cells was found to be $3.2 \times 10^4$ mL$^{-1}$, via PI staining. Thus, the ratio between the densities of metabolically active and dead cells in Lotus Pond's vent water was almost 1.7.

### Microbiome composition reconstructed from the assembled metagenome

Quality filtration of the native metagenomic sequence dataset having 31353866 read-pairs resulted in the retention of 31342225 read-pairs (S1 Table in S1 File). Assembly of the quality-filtered dataset yielded 286918 contigs having a minimum and an average length of 200 and 563 nucleotides respectively (96% of all quality-filtered reads participated in the assembly, at an average coverage of ~85X). While the largest contig obtained was 197270-nucleotide long, the consensus sequence of the assembled metagenome was ~162 million nucleotides, with a total of 379412, complete or partial, ORFs or genes identified therein. Out of the 379412 ORFs identified, 220687 could be annotated as CDSs, of which again 220635 could be ascribed to potential taxonomic sources (S2 Table in S1 File). Approximately 97.3%, 1%, 1.5% and 0.2% of the taxonomically classifiable CDSs, i.e. 214652, 2309, 3276 and 398 CDSs, belonged to Bacteria, Archaea, Eukarya and Viruses respectively (S2 and S3 Tables in S1 File).

The 214652 bacterial CDSs identified were found to be distributed over 23 phyla, with most CDSs being affiliated to Proteobacteria / Pseudomonadota (90793), followed by Aquificae / Aquificota (54151), Firmicutes / Bacillota (18670), Deinococcus-Thermus / Deinococcota (10591), Cyanobacteria (5827), Chloroflexi / Chloroflexota (4254), Bacteroidetes / Bacteroidota (2753), Thermotogae / Thermotogota (2671), Thermodesulfobacteria / Thermodesulfobacteriota (2547), Actinobacteria / Actinomycetota (2357). Unclassified Bacteria accounted for 18386 CDSs, while 13 other (minor) phyla, namely Planctomycetes / Planctomycetota (355), Synergistetes / Synergistota (200), Verrucomicrobia / Verrucomicrobiota (160), Acidobacteria /

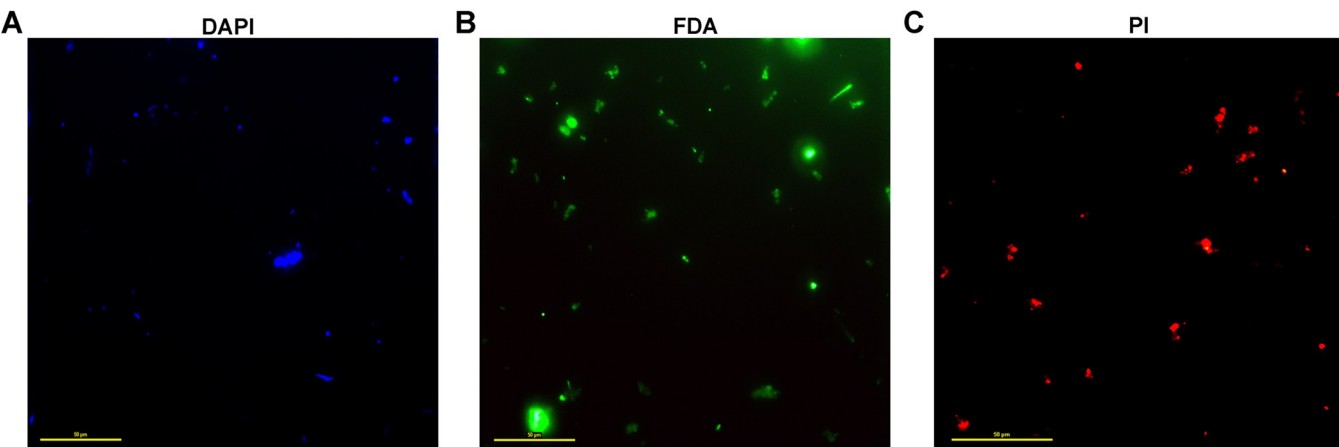

**Fig 2.** Representative picture of the fluorescence microscopic fields based on which microbial cell density was calculated in the vent-water sample of Lotus Pond: (**A**) sample stained with DAPI, (**B**) sample stained with FDA, (**C**) sample stained with PI. Scale bars in all the three micrographs indicate 50 μm length.

Acidobacteriota (134), Fusobacteria / Fusobacteriota (133), Nitrospirae / Nitrospirota (126), Chlorobi / Chlorobiota (93), Spirochaetes / Spirochaetota (77), Thermomicrobia / Thermomicrobiota (63), Deferribacteres / Deferribacterota (43), Tenericutes / Mycoplasmatota (25), Gemmatimonadetes / Gemmatimonadota (21) and Chlamydiae / Chlamydiota (20) were represented sparsely in the CDS catalog (S2 and S3 Tables in S1 File; Fig 3A).

Among the different classes of Proteobacteria, Alphaproteobacteria, followed by Gammaproteobacteria, Betaproteobacteria, Deltaproteobacteria and Epsilonproteobacteria accounted for most CDSs in the assembled metagenome (41306, 31865, 14394, 1145 and 466 CDSs respectively). The other proteobacterial classes Hydrogenophilia, Acidithiobacillia and Oligoflexia were represented by 118, 106 and 101 CDSs respectively, while unclassified Proteobacteria accounted for 1046 CDSs (Fig 3A). Furthermore, within the proteobacterial CDS catalog, the Xanthomonadales-Lysobacterales complex, Vibrionales, Oceanospirillales (all three belonged to the Gammaproteobacteria), and Rhodocyclales (belonging to Betaproteobacteria) were the maximally represented orders, with 10039, 6025, 3,940 and 3582 CDSs affiliated to them respectively (S2 and S3 Tables in S1 File).

The small archaeal genetic diversity was contributed by Crenarchaeota / Thermoproteota, followed by Euryarchaeota (1479 and 647 CDSs respectively); unclassified Archaea accounted for 183 CDSs (S2 and S3 Tables in S1 File). While the Crenarchaeota related CDSs were not classifiable further, the Euryarchaeota-affiliated CDSs were distributed over the classes Halobacteria (177), Thermococci (169), Archaeoglobi (83), Methanococci (60), Methanomicrobia (30), Methanobacteria (19) and Thermoplasmata (9); unclassified Euryarchaeota accounted for 100 CDSs (S2 and S3 Tables in S1 File). Most of the putatively eukaryotic CDSs detected exhibited highest sequence similarities with homologs from Opisthokonta, affiliated to either Fungi or Metazoa (S2 Table in S1 File). The few viral CDSs detected were mostly contributed by Podoviridae, Myoviridae, Siphoviridae and Caudovirales (S2 Table in S1 File).

## Genus-level constitution of the microbiome

Taxonomic classification of CDSs predicted within the assembled metagenome was not extended up to the genus level. Instead, generic components of the microbiome were delineated via taxonomic classification of (i) metagenomic reads corresponding to 16S rRNA genes, and (ii) OTUs clustered from PCR-amplified 16S rRNA gene sequences. Additionally, the

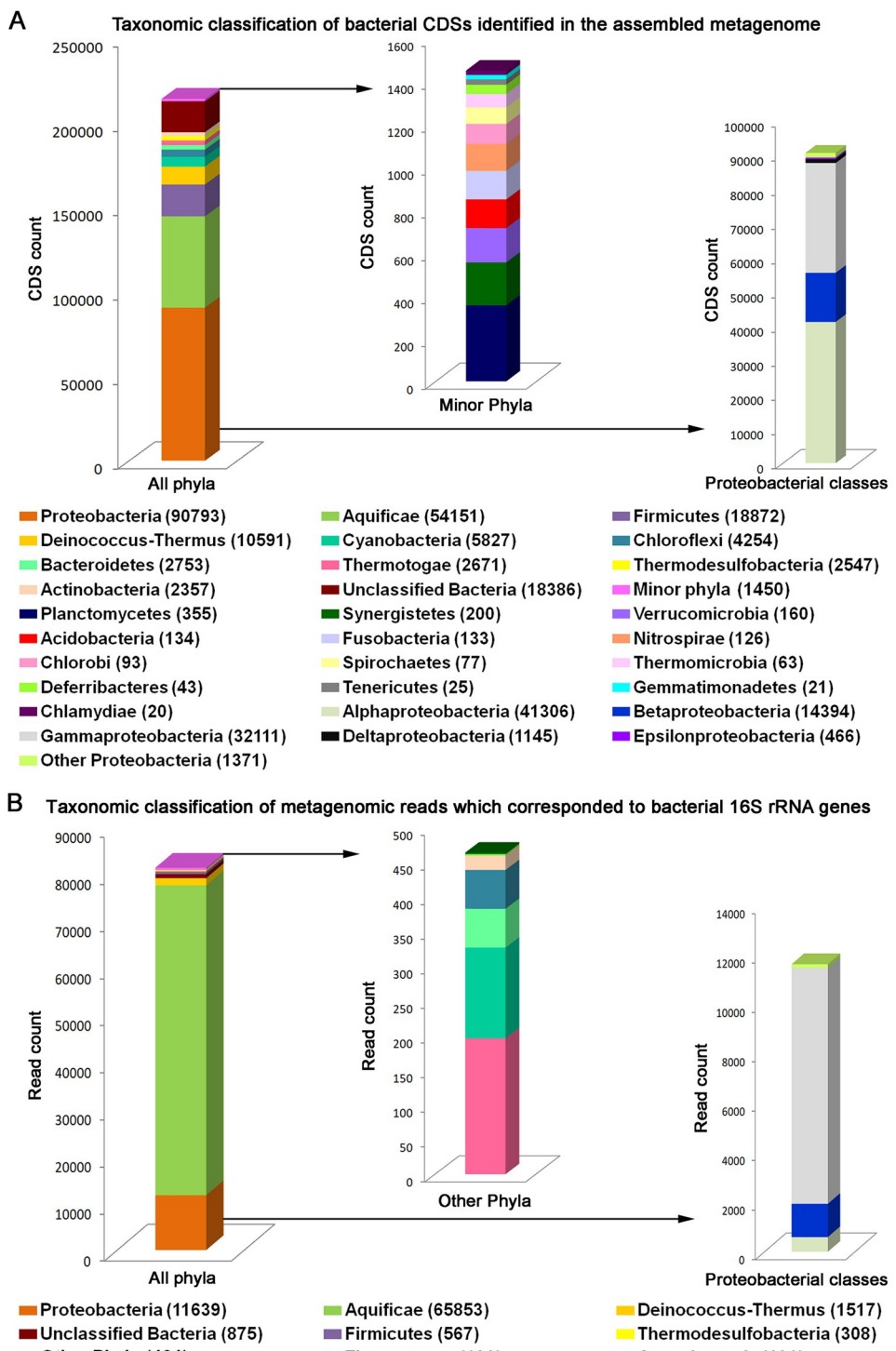

**Fig 3.** Taxonomic distribution of (**A**) the CDSs identified in the assembled metagenome of Lotus Pond, and (**B**) the metagenomic reads which corresponded to 16S rRNA genes (actual CDS count or metagenomic read count recorded for each taxon is given in parenthesis). In (**A**) "minor phyla" included the following: Acidobacteria, Chlamydiae, Chlorobi, Deferribacteres, Fusobacteria, Gemmatimonadetes, Nitrospirae, Planctomycetes, Spirochaetes, Synergistetes, Tenericutes, Thermomicrobia and Verrucomicrobia. In (**B**) "other phyla", also marginal in representation, included

the following: Cyanobacteria, Thermotogae, Bacteroidetes, Chloroflexi, Actinobacteria, Armatimonadetes and Synergistetes.

taxonomic diversity data obtained from the current analysis of PCR-amplified 16S rRNA gene sequences were utilized to evaluate the consistency of the Lotus Pond vent-water microbiome by comparing with equivalent data available from previous explorations of the habitat.

According to the RDP-based classification of the metagenomic reads, out of the 62684450 high quality reads analyzed, 81619 represented parts of prokaryotic 16S rRNA genes: 81223 bacterial and 396 archaeal (S4 Table in S1 File).

Out of the total 81223 metagenomic reads corresponding to bacterial 16S rRNA genes 80348 were classifiable under 12 phyla (the remaining 875 reads belonged to unclassified Bacteria), of which Aquificae accounted for an overwhelming majority (65853) of reads (S4 Table in S1 File; Fig 3B). Substantive number of reads were also ascribable to Proteobacteria (11639), Deinococcus-Thermus (1517) and Firmicutes (567), while Thermodesulfobacteria (308), Cyanobacteria (131), Thermotogae (196), Bacteroidetes (56), Chloroflexi (56), Actinobacteria (21), Armatimonadetes / Armatimonadota (2), and Synergistetes (2) contributed much fewer reads. The proteobacterial reads, again, were mostly from Gammaproteobacteria (9553), followed by Betaproteobacteria (1345) and Alphaproteobacteria (595). Furthermore, 70729 out of the total 81223 bacterial 16S rRNA gene sequence related reads got ascribed to 92 genera (S4 Table in S1 File) led by *Hydrogenobacter* (44547), *Sulfurihydrogenibium* (16371), *Halomonas* (3075), *Vibrio* (2549), *Thermus* (1454), *Tepidimonas* (965) and *Paracoccus* (196). Whereas the two Aquificales genera *Hydrogenobacter* and *Sulfurihydrogenibium*, affiliated to the families Aquificaceae and Hydrogenothermaceae respectively, accounted for 75.8% of all 16S rRNA gene related reads present in the quality-filtered metagenomic sequence dataset; the other Aquificae members detected, namely *Thermocrinis*, *Venenivibrio*, *Persephonella* and *Hydrogenothermus*, accounted for only 180, 48, 6 and 2 such reads respectively (S4 Table in S1 File).

As for the 396 reads corresponding to archaeal 16S rRNA genes, all were attributable to the phylum Crenarchaeota and only 9 were classifiable up to the level of the genus, detecting *Aeropyrum* (6), *Stetteria* (2) and *Pyrobaculum* (1) in the process (S4 Table in S1 File).

A corroborative experiment involving PCR amplification, seuencing and analysis of the V3 regions of all Bacteria-specific 16S rRNA genes present in the metagenome extracted from the hot water discharged by Lotus Pond revealed 602 bacterial OTUs that in turn were classifiable under 16 different phyla: Proteobacteria (352), Firmicutes (63), Bacteroidetes (52), Actinobacteria (40), Cyanobacteria (20), Deinococcus-Thermus (16), Thermotogae (8), Aquificae (3), Acidobacteria (2), Armatimonadetes (2), Chloroflexi (2), Spirochaetes / Spirochaetota (2), Thermodesulfobacteria (2), Dictyoglomi / Dictyoglomota (1), Rhodothermota (1) and Synergistetes (1); unclassified Bacteria accounted for 35 OTUs (S5 Table in S1 File). Of the total 602 bacterial OTUs identified, 249 were classifiable upto the level of genus. Overall, 121 genera were identified in that way (S5 Table in S1 File), with maximum number of OTUs getting affiliated to *Vibrio* (21), followed by *Halomonas* (16) and *Thermus* (13).

Concurrent to the above experiment, PCR amplification, seuencing and analysis of the V4-V5 regions of all Archaea-specific 16S rRNA genes present in the Lotus Pond vent-water metagenome were carried out to reveal the *in situ* archaeal diversity (S5 Table in S1 File). Only 30 OTUs were detected in this way, of which, 10 were ascribable to Crenarchaeota, 6 each belonged to *Euryarchaeota* and Nitrososphaerota, and 8 were not classifiable below the domain level (S5 Table in S1 File). Furthermore, none of the Crenarchaeota OTUs were classifiable at the genus level, whereas out of the 6 OTUs affiliated to Euryarchaeota, 1 and 3 were

classifiable as species of *Methanomassiliicoccus* and *Methanospirillum* respectively; all the 6 Nitrososphaerota OTUs were classified as members of *Nitrososphaera* (S5 Table in S1 File).

## Predominant generic entities identified via population genome binning

When all the >2500 nucleotide contigs present in the assembled metagenome were binned using Metabat2, MaxBin2, and CONCOCT via separate *in silico* experiments, 19, 14 and 30 MAGs were obtained respectively. Refinement and optimization of these 63 population genome bins using DASTool yielded 14 bacterial (Table 1 and S6 Table in S1 File; Fig 4) and one archaeal (Table 1 and S6 Table in S1 File) MAGs for subsequent consideration and analysis. Each of these draft genomes had a contamination level of <5%, except the one designated as LotusPond_MAG_Paracoccus_sp._1. Likewise, most of the short-listed MAGs (9 out of 15) had >90% completeness; two, identified as Paracoccus_sp._2 and Unclassified_Thermodesulfobacteriaceae, had <50% completeness; and four, identified as Paracoccus_sp._1, Fervidobacterium_sp., Thermus_sp. and Unclassified_Bacteria, possessed approximately 51.2%, 70.2%, 70.3% and 81.9% completeness respectively. In terms of consensus sequence length, the population genomes classified as *Vibrio metschnikovii* and *Halomonas* sp. were the largest (approximately 3.5 and 3.4 mb respectively) whereas the one classified as a novel member of the family Thermodesulfobacteriaceae was the

**Table 1. Key features of the population genome bins (MAGs) constructed from the assembled metagenome of Lotus Pond's vent-water.**

| Name of the population genome bin obtained (genome accession number) | % of quality-filtered metagenomic read-pairs matching sequences of the MAG concordantly | No. of contigs | Genome size (nucleotides) | G+C content (%) | Completeness (%) | Contamination (%) | Number of ORFs (genes) / CDSs detected in the MAG |
|---|---|---|---|---|---|---|---|
| LotusPond_MAG_Unclassified_Aquificaceae (JARJNB000000000) | 55.92 | 131 | 1489711 | 43.32 | 95.12 | 0.61 | 1754 / 1707 |
| LotusPond_MAG_Sulfurihydrogenibium_azorense (JARJNA000000000) | 3.48 | 47 | 1516769 | 32.77 | 98.98 | 0.00 | 1680 / 1639 |
| LotusPond_MAG_Halomonas_sp. (JARJMQ000000000) | 2.65 | 97 | 3349510 | 60.21 | 99.46 | 0.94 | 3245 / 3176 |
| LotusPond_MAG_Unclassified_Desulfurococcales (JARJMZ000000000) | 1.60 | 37 | 1553981 | 49.38 | 99.63 | 0.00 | 1621 / 1559 |
| LotusPond_MAG_Vibrio_metschnikovii (JARJMP000000000) | 1.54 | 249 | 3487509 | 44.28 | 94.07 | 1.85 | 3222 / 3172 |
| LotusPond_MAG_Thermus_sp. (JARJMW000000000) | 1.01 | 372 | 1887946 | 62.80 | 70.34 | 4.24 | 2206 / 2157 |
| LotusPond_MAG_Tepidimonas_sp. (JARJMT000000000) | 0.46 | 200 | 2150705 | 66.30 | 94.16 | 1.29 | 2200 / 2155 |
| LotusPond_MAG_Thermosynechococcus_sp. (JARJMS000000000) | 0.35 | 104 | 2328058 | 52.52 | 97.05 | 0.12 | 2367 / 2319 |
| LotusPond_MAG_Tepidimonas_taiwanensis (JARJMR000000000) | 0.28 | 245 | 2568138 | 69.51 | 90.53 | 3.56 | 2567 / 2525 |
| LotusPond_MAG_Fervidobacterium_sp. (JARJMY000000000) | 0.24 | 293 | 1525481 | 39.93 | 70.18 | 0.88 | 1639 / 1611 |
| LotusPond_MAG_Paracoccus_sp._1 (JARJMU000000000) | 0.17 | 528 | 2138329 | 62.94 | 51.22 | 15.08 | 2383 / 2345 |
| LotusPond_MAG_Unclassified_Bacteria (JARJMV000000000) | 0.11 | 335 | 2025372 | 61.84 | 81.86 | 0.31 | 2082 / 2039 |
| LotusPond_MAG_Unclassified_Chromatiales (JARJMX000000000) | 0.10 | 214 | 1644354 | 62.57 | 91.04 | 2.01 | 1796 / 1758 |
| LotusPond_MAG_Unclassified_Thermodesulfobacteriaceae (JARJND000000000) | 0.07 | 114 | 439694 | 36.23 | 30.15 | 0.31 | 477 / 467 |
| LotusPond_MAG_Paracoccus_sp._2 (JARJNC000000000) | 0.06 | 284 | 1087115 | 64.14 | 34.31 | 1.10 | 1214 / 1201 |

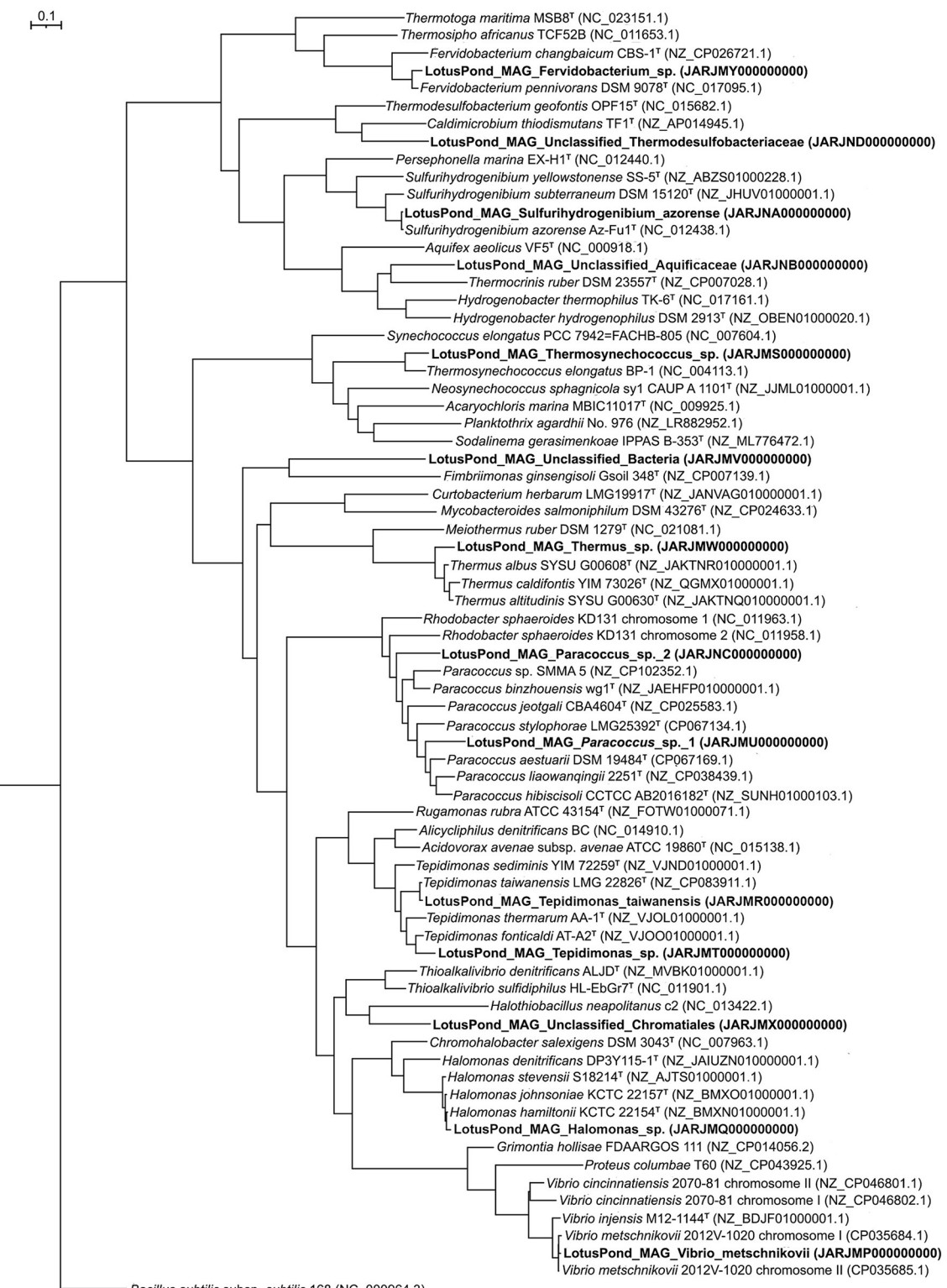

**Fig 4. Phylogenetic tree (number of bootstrap tests carried out = 10000) based on 92 universal bacterial core genes that shows the evolutionary relationships among the 14 bacterial MAGs reconstructed from the Lotus Pond vent-water metagenome and their closest relatives identified based on dDDH and/or rRNA gene sequence relationships.** Scale bar denotes a distance equivalent to 10% nucleotide substitution. In the phylogeny reconstructed, nucleotide substitutions were interpreted using the generalized time-reversible model, which considers the proportion of invariable sites and/or the rate of variation across the sites.

smallest (approximately 0.4 mb only). Furthermore, when the 31342225 high quality read-pairs used for metagenome assembly were searched against the 15 population genomes, a total of 68.08% read-pairs mapped concordantly on to the MAGs as a whole. Individually, the 15 MAGs accounted for 0.06% to 55.9% of the metagenomic read-pairs analyzed.

The genome bin LotusPond_MAG_Unclassified_Aquificaceae accounted for an overwhelming proportion of the metagenome (55.9% of all read-pairs assembled mapped back to this MAG), while another Aquificales bin identified as *Sulfurihydrogenibium azorense*, a member of Hydrogenothermaceae, accounted for 3.5% of all read-pairs. Four more bins individually represented ≥1% of the metagenome; these were identifed as (i) an unknown species of the genus *Halomonas*, (ii) an unclassified member of the order Desulfurococcales within the archaeal class Thermoprotei, (iii) the gammaproteobacteium *Vibrio metschnikovii*, and (iv) an unknown species of the genus *Thermus*. These MAGs accounted for 2.7%, 1.6%, 1.5% and 1% of all metagenomic read-pairs respectively. Seven other MAGs, ascribed to *Tepidimonas taiwanensis*, *Tepidimonas* sp., *Fervidobacterium* sp., *Thermosynechococcus* sp., *Paracoccus* sp., an unclassified member of the gammaproteobacterial order Chromatiales, and an unclassified taxon of Bacteria, individually accounted for >0.1% but <1% of the metagenome. The remaining two Lotus Pond MAGs individually represented <0.1% of the metagenome; these were identified as one unclassified member each from the family Thermodesulfobacteriaceae of the monotypic phylum Thermodesulfobacteria, and the genus *Paracoccus*. Details of the phylogenomic relationships, and the consequent basis of taxonomic identification, of the individual MAGs are given in Supplementary Results.

## Environmentally advantageous functions encoded by the Lotus Pond MAGs

Apart from house-keeping genes, all the MAGs encompassed genes for heat stress management. Irrespective of their completeness level, MAGs affiliated to thermophilic taxa such as Aquificaceae, *Sulfurihydrogenibium*, Desulfurococcales, Thermodesulfobacteriaceae and *Thermus* had less than 10 such genes each. In contrast, almost all the MAGs affiliated to mesophilic taxa had ≥10 heat stress management genes each. Almost all the MAGs contained genes for managing oxidative and periplasmic stress; a half of them had genes for withstanding carbon starvation and nitrosative stress. Eight MAGs encoded genes conferring high frequency of lysogeny as a means of stress response. Every bacterial MAG, but not the archaeal one, encompassed genes for stringent response to environmental stimuli via (p)ppGpp metabolism. Most of the MAGs encoded versatile sporulation-associated proteins, while those affiliated to mesophilic or moderately thermophilic taxa encoded a universal stress protein. Four of the 15 MAGs possessed 12–34 genes for different toxin-antitoxin systems, eight had 1–9 such genes, whereas three (belonging to Aquificaceae, Desulfurococcales and Thermodesulfobacteriaceae) had none. All but one MAG contained genes for cation transport; most had genes for twin-arginine motif-based protein translocation and Ton and Tol transport systems, while some encoded ABC transporters, uni- sym- and antiporters, TRAP transporters, and various secretion systems. Half of the MAGs harbored genes for flagellar motility or chemotaxis; protection against metals, toxins, and antibiotics; and CRISPR systems.

Details of the adaptationally useful genes encompassed by the different MAGs are given in Supplementary Results.

## Microbiome functions predicted from the assembled metagenome

Out of the 379412 genes predicted within the assembled Lotus Pond metagenome by Prodigal v2.6.3 212749 were ascribable to COGs that again were distributed across 25 functional

categories (S7 Table in S1 File). Of these, Translation, ribosomal structure and biogenesis; and Energy production and conversion had the maximum number of CDSs ascribed to them (26153 and 22448 respectively). Next in attendance were the COG categories Amino acid transport and metabolism (19711 CDSs); Cell wall/membrane/envelope biogenesis (15685 CDSs); Coenzyme transport and metabolism (14510 CDSs); and Inorganic ion transport and metabolism (11949 CDSs). Notably, the lowest numbers of CDSs, i.e. 26, 23, 21 and 3, were allocated to Extracellular structures; Cytoskeleton; RNA processing and modification; and Chromatin structure and dynamics, respectively. The COG category termed as Nuclear structure had no gene assigned to it at all (S7 Table in S1 File).

For energy production and conversion, maximum homologs were detected for the genes encoding NADH-quinone oxidoreductase and cytochrome *c* oxidase. While oxidative phosphorylation was the most represented pathway under energy production and conversion, NADH:quinone oxidoreductase, the reductive tricarboxylic acid (rTCA) cycle or the reductive citrate cycle, the TCA or citrate cycle, and cytochrome *c* oxidase were the biochemical modules with most CDSs (S2 Table in S1 File).

For DNA metabolism, maximum number of homologs was detected for a transposase encoding gene. Mismatch repair, homologous recombination, replication, and nucleotide excision repair were the most represented pathways, while DNA polymerase III complex and fluoroquinolone-resistance gyrase-protecting protein Qnr, were the most represented biochemical modules (S2 Table in S1 File).

For translation and ribosomal structure and biogenesis, maximum homologs were detected for the elongation factor Tu. Aminoacyl-tRNA biosynthesis was the most represented pathway, while ribosome (bacterial followed by archaeal) was the maximally represented biochemical module (S2 Table in S1 File).

Regarding amino acid transport and metabolism, maximum homologs were there for glycine cleavage system protein H and cysteine desulfurase. Biosynthesis of secondary metabolites was the most represented pathway, while maximum CDSs were attributed to the biochemical module photorespiration.

As for cell wall/membrane/envelope biogenesis, maximum homologs were detected for the transaminase which isomerizes glutamine:fructose-6-phosphate, and the GTP-binding protein LepA, which acts as a potential fidelity factor of translation (for accurate and efficient protein synthesis) under certain stress conditions [63,64]. Under this category, amino sugar and nucleotide sugar metabolism, antibiotics biosynthesis, and peptidoglycan biosynthesis were the most represented pathways. At the module level, nucleotide sugar biosynthesis, dTDP-L-rhamnose biosynthesis, lipoprotein-releasing system, lipopolysaccharide biosynthesis (KDO2-lipid A), and multidrug resistance related efflux pumps AcrAB-TolC/SmeDEF, AcrAD-TolC, MexAB-OprM and AmeABC were maximally represented (S2 Table in S1 File).

For coenzyme transport and metabolism, maximum homologs were detected for uroporphyrin-III C-methyltransferase. Secondary metabolites biosynthesis, porphyrin metabolism, pantothenate and co-enzyme A (CoA) biosynthesis, and antibiotics biosynthesis were the most represented pathways, while heme biosynthesis was the most represented biochemical module (S2 Table in S1 File).

Regarding inorganic ion transport and metabolism, maximum homologs were detected for NADH-quinone oxidoreductase subunit L. Under this category, ABC transporters, oxidative phosphorylation, quorum sensing, and two-component system, were the most represented pathways, wheras at the module level, NADH:quinone oxidoreductase, peptides/nickel transport system, phosphate transport system, iron complex transport system, nickel transport system, NitT/TauT family transport system, nitrate/nitrite transport system, and iron(III) transport system were represented maximally (S2 Table in S1 File).

## Chemolithotrophic sulfur oxidation (Sox) is the primary bioenergetic mechanism of the Lotus Pond vent-water ecosystem

Out of all the mechanisms of chemolithotrophy known, only sulfur oxidation (Sox) was found to have substantial numbers of coding sequences, partial or complete, in the assembled meta-genome for most of its constituent biochemical reactions (enzymatic steps). Overall, 333 CDSs were there for the different components of the Sox multienzyme complex: 21 for the L-cysteine S-thiosulfotransferase protein SoxX (K17223), 53 for the SoxA subunit of L-cysteine S-thiosul-fotransferase (K17222), 112 and 57 for the sulfur compounds chelating/binding protein sub-units SoxY (K17226) and SoxZ (K17227) respectively, 79 for the S-sulfosulfanyl-L-cysteine sulfohydrolase SoxB (K17224), and 11 for the sulfane dehydrogenase protein's molybdopterin-containing subunit SoxC (K17225) and diheme *c*-type cytochrome subunit SoxD (K22622), the CDSs of which occur universally as overlapping ORFs. In terms of taxonomic distribution, maximum number of Sox protein encoding sequences (~40% of the total 333) was ascribed to Aquificae, followed by Alphaproteobacteria (~25%), Betaproteobacteria (14%), and Deinococ-cus-Thermus (9%). Unclassified Bacteria, Gammaproteobacteria, and other Proteobacteria individually accounted for approximately 8, 4 and 1 percent of all the Sox CDSs annotated (S2 Table in S1 File).

As for the other forms of chemolithotrophy known in the microbial world, only a small number of CDSs could be detected for the key proteins of arsenite and hydrogen oxidation. Whereas five CDSs each were detected for the small and large subunits AoxA (K08355) and AoxB (K08356) of arsenite oxidase respectively, only a single CDS was identifiable for the small subunit of ferredoxin hydrogenase (K00534), even as no CDS could be detected for the hydrogenase large subunit (K00533). Notably, no CDS was also there for any of the other hydrogenase variants known in the literature.

No CDS was present in the Lotus Pond microbiome for the chemolithotrophic iron oxida-tion protein iron:rusticyanin reductase (K20150). Likewise, there was also no CDS for the enzymes concerned with the chemolithotrophic oxidation of ammonia to nitrite. However, despite the absence of genes encoding the different subunits of methane/ammonia monooxy-genase (*pmoABC*—K10944, K10945 and K10946), or for that matter the nonexistence of any CDS for the other ammonia-oxidizing enzyme hydroxylamine dehydrogenase (K10535), the Lotus Pond metagenome did encompass several genes for the alpha (K00370) and beta (K00371) subunits of nitrite to nitrate oxidizing enzyme nitrate reductase / nitrite oxidoreduc-tase (269 and 90 CDSs respectively).

So far as the population genome bins are concenrned, the one designated as LotusPond_-MAG_Unclassified_Aquificaceae was found to have genes coding for the proteins SoxX, SoxA, SoxB, SoxY, and SoxZ, plus the accessory protein SoxW, which is a thioredoxin serving the purpose of reducing the different subunits of the Sox multienzyme complex to their function-ally active states. The genome bin designated as LotusPond_MAG_Thermus_sp. encoded the proteins SoxX, SoxA, SoxB, SoxC, and SoxD, while the one entitled as LotusPond_MAG_Un-classified_Chromatiales contained genes for SoxX, SoxA, SoxB, and SoxC. LotusPond_MAG_-Paracoccus_sp._1 contained gene encoding SoxB, SoxCD, and the additional sulfur oxidation proteins SoxF (a periplasmic flavoprotein sulfide dehydrogenase) and SoxG (a thiosulfate-induced periplasmic zinc metallohydrolase that can act as a potential thiol esterase), while the largely incomplete genome bin designated as LotusPond_MAG_Paracoccus_sp._2 contained genes for only SoxY, SoxZ, and the transcriptional regulator of the *sox* operon called SoxR.

The population genome bin LotusPond_MAG_Sulfurihydrogenibium_azorense encom-passed genes for all the four subunits—alpha (K17993), delta (K17994), gamma (K17995) and beta (K17996)–of the signature protein of the genus called sulfhydrogenase (HydADGB),

which is $H_2$:polysulfide oxidoreductase catalyzing hydrogen oxidation via reduction of elemental sulfur or polysulfide to hydrogen sulfide.

## Centrality of the rTCA cycle, and Aquificae, in Lotus Pond's primary productivity

In the assembled metagenome of Lotus Pond there were substantial numbers of CDSs, whether complete or partial, for all the enzymatic steps necessary to execute (i) the rTCA (Arnon-Buchanan) as well as the modified rTCA cycle [65,66], (ii) the reductive pentose phosphate (Calvin-Benson-Bassham) cycle [67], (iii) the 3-hydroxypropionate bi-cycle [68], and (iv) the reductive acetyl Co-A (Wood-Ljungdahl) pathway [65,69]. For the other major pathways of carbon assimilation (autotrophic carbon fixation), namely (v) the dicarboxylate/ 4-hydroxybutyrate cycle [70] and (vi) the 3-hydroxypropionate/4-hydroxybutyrate cycle [65], genes concerned with a number of intermediate transformations were either missing from the metagenome or were represented by just one or two CDSs (Fig 5). For the other recently-reported and not so common pathways, the key biochemical steps are rendered by the specialized activities of certain typical enzyme variants for which genes are difficult to identify on the basis of homology analysis alone. These pathways are (vii) the reverse oxidative TCA (roTCA) cycle, where citrate is converted to acetyl-CoA and oxaloacetate by a reverse acting citrate synthase homolog [71], (vii) the transaldolase variant of the Calvin cycle mediated by Form III RubisCO [72], and (ix) the reductive glycine pathway, where $CO_2$ is converted to formate by the action of a formate dehydrogenase, and 5,10-methylene-tetrahydrofolate is converted to glycine by a glycine cleavage/synthase system [73].

In case of rTCA cycle, CDSs detected in Lotus Pond's assembled metagenome included 5, 218, and 57 homologs for (i) ATP-citrate lyase AclAB (K15230 and K15231), (ii) NADH-dependent fumarate reductase FrdABCDE (K18556, K18557, K18558, K18559 and K18560), and (iii) ferredoxin-dependent 2-oxoglutarate synthase/oxidoreductase KorABCD (K00174, K00175, K00176 and K00177), which catalyze the key reactions (i) citrate to acetyl-CoA and oxaloacetate, (ii) fumarate to succinate, and (iii) succinyl CoA to 2-oxoglutarate, respectively. Most of these CDSs exhibited closest sequence similarities with homologs from members of Aquificae followed by Thermotogae. CDSs corresponding to K18556, K18557, K18558, K18559 and K18560 were also detectable within LotusPond_MAG_Unclassified_Aquificaceae, while CDSs corresponding to K15230 and K15231, and K18556 and K18557 were identified within LotusPond_MAG_Sulfurihydrogenibium_azorense. Furthermore, in the assembled metagenome of Lotus Pond, 411, 34, 57 and 274 CDSs were detected for the conversion of (i) acetyl-CoA to pyruvate (with $CO_2$ addition), (ii) phosphoenolpyruvate to oxaloacetate (with $HCO_3^-$ addition), (iii) succinyl-CoA to 2-oxoglutarate, and (iv) 2-oxoglutarate to isocitrate (the last two conversions involve $CO_2$ addition) respectively (Fig 5A).

Apart from the key rTCA cycle genes mentioned above, 108 and 112 CDSs were detectable in Lotus Pond's assembled metagenome for the two subunits of citryl-CoA synthetase (K15232 and K15233 respectively), while two CDSs were also there for citryl-CoA lyase (K15234). Notably, all the CDSs corresponding to these two enzymes catalyzing the two key reactions of the modified rTCA cycle, namely citrate to citryl-CoA, and citryl-CoA to oxaloacetate and acetyl-CoA, exhibited closest sequence similarities with homologs from Aquificae. Corroboratively, CDSs corresponding to K15232, K15233 and K15234 were identified within LotusPond_MAG_Unclassified_Aquificaceae, while a single CDS corresponding to K15234 was detectable in LotusPond_MAG_Sulfurihydrogenibium_azorense.

As regards the reductive pentose phosphate cycle, CDSs detected in Lotus Pond's assembled metagenome included 32 and 40 homologs for (i) phosphoribulokinase (K00855) and (ii)

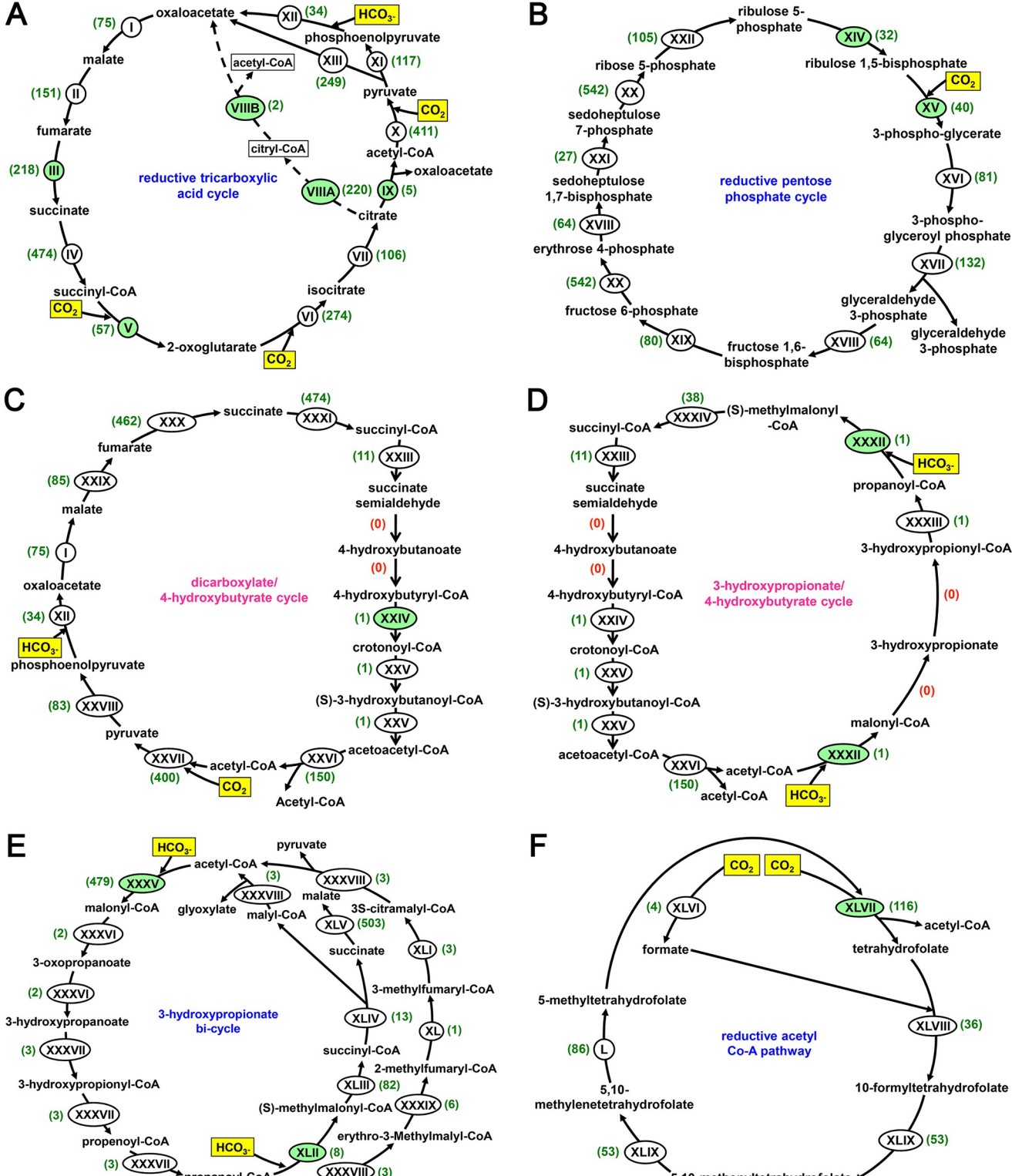

**Fig 5. Simplified schematic view of the carbon assimilation pathways for which genes were identified in the assembled metagenome of Lotus Pond.** The relevant genes were identified based on the annotation of CDSs with reference to the published literature, and biochemical modules curated under the KEGG Pathway Maps database located at https://www.genome.jp/kegg/pathway.html: (**A**) reductive citrate cycle, (**B**) reductive pentose phosphate cycle, (**C**) dicarboxylate/4-hydroxybutyrate cycle, (**D**) 3-hydroxypropionate/4-hydroxybutyrate cycle, (**E**) 3-hydroxypropionate bi-cycle, and (**F**) reductive acetyl Co-A pathway. Names of the pathways for which one or more genes were undetectable in the metagenome have been written in pink fonts. The active

inorganic carbon substrates assimilated by the different pathways are highlighted in yellow. The biochemical conversions (steps) of the different pathways for which CDSs could be identified in the assembled metagenome are indicated by the roman numerals I through L. While the roman numerals shaded green indicate the key enzymatic steps of the pathway in question, the numbers given in parentheses denote the numbers of CDSs identified for the individual enzymatic steps. The enzyme-coding genes actually detected in the metagenome for the biochemical steps labeled as I through L are enumerated below by their KEGG orthology identifiers. I—K00024; II—K01676, K01677, K01678, K01679; III—K18556, K18557, K18558, K18559, K18560; IV—K01902, K01903; V—K00174, K00175, K00176, K00177; VI—K00031; VII—K01681, K01682; VIIIA—K15232, K15233; VIIB—K15234; IX—K15230, K15231; X—K00169, K00170, K00171, K00172, K03737; XI—K01006, K01007; XII—K01595; XIII—K01958, K01959, K01960; XIV—K00855; XV—K01601, K01602; XVI—K00927; XVII—K00134, K00150; XVIII—K01623, K01624; XIX—K01086, K02446, K03841, K11532; XX—K00615; XXI—K01086, K11532; XXII—K01807, K01808; XXIII—K15038; XXIV—K14534; XXV—K15016; XXVI—K00626; XXVII—K00169, K00170, K00171, K00172; XXVIII—K01007; XXIX—K01677, K01678; XXX—K00239, K00240, K00241, K18860; XXXI—K01902, K01903; XXXII—K15037; XXXIII—K15020; XXXIV—K05606, K01848, K01849; XXXV—K02160, K01961, K01962, K01963; XXXVI—K14468; XXXVII—K14469; XXXVIII—K08691; XXXIX—K14449; XL—K14470; XLI—K09709; XLII—K15052; XLIII—K05606, K01847, K01848, K01849; XLIV—K14471, K14472; XLV—K00239, K00240, K00241, K01679; XLVI—K05299, K22015; XLVII—K00194, K00195, K00196, K00197, K00198, K14138; XLVIII—K01938; XLIX—K01491; L—K00297. In the panel A showing the different steps of the rTCA cycle, dotted lines indicate the reactions of the modified rTCA cycle which are mediated by the enzymes citryl-CoA synthetase (VIIIA) and citryl-CoA lyase (VIIIB).

ribulose-1,5-bisphosphate carboxylase/oxygenase (RubisCO, K01601 and K01602), which catalyze the key reactions (i) ribulose-5-phosphate to ribulose-1,5-bisphosphate, and (ii) ribulose-1,5-bisphosphate to 3-phosphoglycerate (the sole $CO_2$ fixing step of the pathway), respectively (Fig 5B). While most of these CDSs exhibited closest sequence similarities with homologs from members of Deinococcus-Thermus, CDSs corresponding to K00855, as well as K01601 and K01602, were there in LotusPond_MAG_Thermus_sp.; notably, homologs of K01601 and K01602 encoding the large (RbcL) and small (RbcS) subunits of RubisCO were also detected in LotusPond_MAG_Thermosynechococcus_sp. and LotusPond_MAG_Unclassified_Chromatiales.

So far as the dicarboxylate/4-hydroxybutyrate cycle is concerned, only one Lotus Pond CDS was found for 4-hydroxybutyryl-CoA dehydratase (K14534), which governs the key step of converting 4-hydroxybutyryl-CoA to crotonyl-CoA. Moreover, though 400 and 34 CDSs were there in the Lotus Pond metagenome for the conversion of (i) acetyl-CoA to pyruvate, and (ii) phosphoenolpyruvate to oxaloacetate, via addition of $CO_2$ and $HCO_3^-$ respectively, the numbers of CDSs detected for the reactions of the hydroxybutyrate half of the cycle were extremely low (Fig 5C). In this context it is noteworthy that the carbon fixation reactions of the dicarboxylate/4-hydroxybutyrate cycle are same as the first two carbon fixation steps of the rTCA cycle: phosphoenolpyruvate to oxaloacetate conversion involves the same enzyme phosphoenolpyruvate carboxylase (KEGG orthology: K01595) in either pathway, while there is only one additional enzyme [pyruvate-ferredoxin/flavodoxin oxidoreductase Por (K03737)] for acetyl-CoA to pyruvate conversion by the rTCA cycle, and that precisely accounted for the extra 11 CDSs ascribed to the corresponding step of the rTCA cycle (Fig 5). Evidently, the high CDS counts for the carbon fixation reactions of dicarboxylate/4-hydroxybutyrate cycle were only a reflection of the overall preponderance of the rTCA cycle, and dicarboxylate/4-hydroxybutyrate cycle was plausibly not prevalent in the microbiome.

The 3-hydroxypropionate/4-hydroxybutyrate cycle is characterized by its signature protein acetyl-CoA/propionyl-CoA carboxylase (biotin-dependent). This protein by virtue of its three subunits structured in an $\alpha_4\beta_4\gamma_4$ configuration and having the functional domains for biotin carboxylase (K01964), biotin carboxyl carrier protein (K15037), and carboxytransferase (K15036) respectively, catalyzes (i) the conversion of acetyl-CoA to malonyl-CoA, as well as (ii) the conversion of propanoyl-CoA to (S)-methylmalonyl-CoA, using $HCO_3^-$ as the active inorganic carbon source in both the reactions. The Lotus Pond metagenome had only one Crenarchaeota-affiliated CDS matching the gene K15037 but no CDS for the genes K01964 and K15036 (Fig 5D).

The key steps of the 3-hydroxypropionate bi-cycle involve carbon fixation reactions similar to those encountered in the 3-hydroxypropionate/4-hydroxybutyrate cycle. However, the two

$HCO_3^-$ assimilating reactions in the bi-cycle are catalyzed by enzymes that are distinct from the acetyl-CoA/propionyl-CoA carboxylase of 3-hydroxypropionate/4-hydroxybutyrate cycle. For instance, conversion of acetyl-CoA to malonyl-CoA in the bi-cycle is rendered by a separate acetyl-CoA carboxylase complex AccABCD, which is made up of the biotin carboxylase subunit AccC (K01961), the biotin carboxyl carrier AccB (K02160), and the carboxyl transferase subunits alpha (AccA; K01962) and beta (AccD; K01963). Propanoyl-CoA, in the bi-cycle, is converted to (S)-methylmalonyl-CoA by an independent propionyl-CoA carboxylase (K15052). The Lotus Pond metagenome encompassed 479 CDSs for the AccABCD complex, whereas only eight CDSs could be identified for propionyl-CoA carboxylase (Fig 5E). Most of the Lotus Pond CDSs corresponding to AccABCD and propionyl-CoA carboxylase exhibited closest sequence similarities with homologs from Aquificae and Chloroflexi respectively. The Lotus Pond MAGs affiliated to unclassified Aquificaceae and *Sulfurihydrogenibium azorense* encompassed CDSs for all the subunits of AccABCD.

In the key step of the Wood–Ljungdahl pathway an $O_2$-sensitive bifunctional multienzyme complex called carbon monoxide dehydrogenase/acetyl-CoA synthase (CODH/ACS) first reduces a $CO_2$ molecule to a CO (carbonyl) moiety, and then combines the CO with a methyl ($CH_3$) group (furnished by the other branch of the pathway) and CoA to generate acetyl-CoA. The CODH/ACS complex is made up of five subunits and four of them are homologous across Archaea and Bacteria. In Archaea they are referred to as the α, β, γ and δ subunits, or the CdhA (K00192), CdhC (K00193), CdhE (K00197) and CdhD (K00194) proteins, whereas in Bacteria the same four homologs are known as the β, α, γ and δ subunits, or the AcsA (K00198), AcsB (K14138), AcsC (K00197) and AcsD (K00194) proteins, respectively. In addition, an ε subunit called CdhB (K00195) is exclusively present in the Archaea, while the AcsE (K15023) protein is exclusive to Bacteria. Overall 13 CDSs for the different components of the Cdh and/or Acs complex were identified in Lotus Pond's assembled metagenome (Fig 5F), and most of them showed closest sequence similarities with homologs from Thermodesulfobacteria. Concurrently, LotusPond_MAG_Unclassified_Thermodesulfobacteriaceae encompassed one CDS each for AcsA, AcsB and AcsD.

## Copious invasion, virulence, and antibiosis related genes in the Lotus Pond metagenome

While there were several complete or partial CDSs (a total of 76) in the Lotus Pond ventwater's assembled metagenome for proteins concerned with biofilm formation and cellular adhesion, quite a few CDSs attributed to hemagglutinin and hemolysin biosynthesis (86), and stationary phase metabolisms including the formation of persister cells (43) were also there (S2 Table in S1 File). Furthermore, the metagenome encompassed 1084 complete or partial CDSs encoding the different components of Type-I, Type-II, Type-III, Type-IV, Type-VI, Type-VIII and Type-IX secretion systems (SSs). Of these, most were parts of the Type-II, and Type-III and Type-IV SSs. In terms of taxonomic distribution (S2 Table in S1 File), most of the SS related CDSs identified (57% of the total 1084) were attributed to Aquificae, followed by Betaproteobacteria (~10%). Gammaproteobacteria and Firmicutes each accounted for 8%, while Alphaproteobacteria accounted for ~5% of all these CDSs. Approximately 1–3% of all SS related CDSs were ascribable to Deinococcus-Thermus, Unclassified Proteobacteria, Unclassified Bacteria, Thermodesulfobacteria, Cyanobacteria, Actinobacteria, and Archaea. Thermotogae accounted for ~0.6% of the SS related CDSs identified in the assembled metagenome. The Lotus Pond MAGs also encompassed quite a few CDSs for biofilm formation, cellular adhesion, invasion and virulence, with the one identified as *Vibrio metschnikovii* having the maximum number and diversity of CDSs for these functions.

Genomic loci (gene clusters) concerned with the biosynthesis of diverse bioactive natural compounds were identified using the antiSMASH pipeline. By searching for specific sequence signatures in the putative biosynthetic gene clusters (BGCs) detected the following classes of secondary metabolites were predicted to be synthesized within the microbiome: N-acyl amino acids, aryl polyene compounds, protease inhibitor-containing beta-lactones, ectoines, homoserine lactones, and terpenes. BGCs corresponding to non-ribosomal peptide synthetases (NRPSs) and NRPS-like fragments, Type I and other polyketide synthases, ribosomally synthesised and post-translationally modified peptides (RiPPs), RiPP recognition elements, and a host of unclassified secondary metabolite-related proteins were also detectable in Lotus Pond's assembled metagenome. Overall, approximately 3000 complete or partial CDSs (S2 and S8 Tables in S1 File) could be identified as parts of BGCs and/or involved directly or indirectly in the biosynthesis of antibiotics such as the acarbose and validamycin, carbapenems, monobactams, neomycin/kanamycin/gentamicin, novobiocin, penicillins and cephalosporins, phenazine, prodigiosin, staurosporine, streptomycin, and vancomycin. So far as the key biosynthetic genes are concerned, several CDSs for non-ribosomal peptide synthetase and prephenate dehydrogenase, which are central to the vancomycin biosynthesis pathway, were found in the Lotus Pond metagenome (S2 Table in S1 File). CDSs for myo-inositol-1-phosphate synthase, dTDP-4-dehydrorhamnose 3,5-epimerase, and dTDP-4-dehydrorhamnose reductase, which are all crucial for the biosynthesis of streptomycin were also detected alongside CDSs for the phenazine biosynthesis protein PhzF (S2 Table in S1 File). For the biosynthesis of carbapenems, several homologs of the *carA*, *carB* and *carC* genes that encode the carbamoyl phosphate synthetase holoenzyme involved in the initial steps of carbapenem biosynthesis (formation of the carbapenem backbone) were detected alongside homologs of *carD* that is involved in the modification of the carbapenem backbone (S2 Table in S1 File). In terms of taxonomic distribution (S8 Table in S1 File), maximum number of these CDSs (~29% of the total) was attributed to Aquificae, followed by Alphaproteobacteria (~21%). The entire domain Archaea accounted for only ~2% of all antibiotic biosynthesis related genes identified. Apart from the BGCs, several genomic loci concerned with the resistance of diverse antibiotics, disinfecting agents, and antiseptics were identified by annotating the assembled metagenome with reference to the CARD database. The antimicrobial agents corresponding to which putative resistance related gene clusters were detected included beta-Lactam compounds, chloramphenicol, erythromycin, fluoroquinolones, metronidazole, teicoplanin, tetracycline, trimethoprim, and vancomycin. In the eggNOG-based annotation (S2 and S8 Tables in S1 File), overall ~5500 such complete or partial CDSs were identified in the assembled metagenome of Lotus Pond's vent-water that were directly or indirectly involved in the resistance of aminoglycosides, bacitracin, beta-lactam compounds, cationic antimicrobial peptides, fluoroquinolones, imipenem, nisin, polymyxin antibiotics, tetracycline, and vancomycin. Furthermore, a wide variety of multi-drug resistance related efflux pumps were detected in the assembled metagenome via eggNOG annotation. In terms of taxonomic distribution (S8 Table in S1 File), considerable number of these antibiotic resistance related CDSs (30% of the total) was attributed to Aquificae, followed by Gammaproteobacteria (~28%). The entire domain Archaea accounted for only one antibiotic resistance related CDS, and the same was meant to protect against aminoglycosides.

## Discussion

### Comparing the microbial diversities revealed by shotgun metagenomics and 16S amplicon sequencing

The extents of Lotus Pond's microbial diversity revealed by the different approaches of molecular genetics had their own differences, as well as concurrence. For instance, two archaeal

phyla, Crenarchaeota and Euryarchaeota, were identified by taxonomic annotation of the CDSs predicted within the assembled metagenome (S3 Table in S1 File), but searching of metagenomic reads for 16S rRNA-encoding sequences detected only Crenarchaeota (S4 Table in S1 File), whereas PCR amplified 16S rRNA gene sequence based OTU analysis identified Crenarchaeota, Euryarchaeota and Nitrososphaerota (S5 Table in S1 File). Concurrently, the archaeal genera *Aeropyrum*, *Pyrobaculum* and *Stetteria*, identified by direct annotation of metagenomic reads, went undetected in 16S amplicon-based analysis, whereas *Methanomassiliicoccus*, *Methanospirillum* and *Nitrososphaera*, detected via 16S amplicon sequencing, were undetectable in the direct annotation of metagenomic reads (compare S4 and S5 Tables in S1 File).

For Bacteria, 23 phyla were identified via taxonomic classification of CDSs: of these, 10 (Actinobacteria, Aquificae, Bacteroidetes, Chloroflexi, Cyanobacteria, Deinococcus-Thermus, Firmicutes, Proteobacteria, Thermodesulfobacteria, and Thermotogae) were major phyla embracing >2000 CDSs each, and 13 (Acidobacteria, Chlamydiae, Chlorobi, Deferribacteres, Fusobacteria, Gemmatimonadetes, Nitrospirae, Planctomycetes, Spirochaetes, Synergistetes, Tenericutes, Thermomicrobia, and Verrucomicrobia) were minor phyla with <500 CDSs each (S3 Table in S1 File; Fig 3A). Annotation of metagenomic reads corresponding to 16S rRNA genes detected 12 phyla in all. These included all the 10 major phyla, and only Synergistetes out of the 13 minor phyla, detected via taxonomic classification of CDSs. Additionally, only one such new phylum, namely Armatimonadetes, was detected via direct metagenomic reads (S4 Table in S1 File; Fig 3B) classification that was not represented in the CDS catalog (S3 Table in S1 File; Fig 3A). 16S amplicon-based OTU analysis identified 16 bacterial phyla (S5 Table in S1 File), which included the 10 major ones identified from CDS annotation, three of the 13 minor phyla (Acidobacteria, Spirochaetes and Synergistetes) detected via CDS annotation, plus Armatimonadetes, and two new phyla (Dictyoglomi and Rhodothermota) that were not represented among the CDSs or the metagenomic reads encoding 16S rRNAs.

Annotation of metagenomic reads corresponding to bacterial 16S rRNA genes revealed 92 bacterial genera (S4 Table in S1 File), while OTU analysis of PCR amplified 16S rRNA gene sequences identified 121 (S5 Table in S1 File). Between these two sets of generic names obtained using two different approaches, 50 genera were common, of which seven (*Fervidobacterium*, *Halomonas*, *Paracoccus*, *Sulfurihydrogenibium*, *Tepidimonas*, *Thermus* and *Vibrio*) were represented in the MAGs obtained from the assembled metagenomic data, and two (*Hydrogenobacter* and *Thiofaba*) were represented by very closely related MAGs (namely, LotusPond_MAG_Unclassified_Aquificaceae and LotusPond_MAG_Unclassified_Chromatiales; detailed phylogenomic relationships are given in the Supplementary Results). Of all the genera represented among the MAGs (Tables 1 and S6 in S1 File; Fig 4), *Thermosynechococcus* alone was not detected in any 16S rRNA gene sequence-based analysis; however, LotusPond_MAG_Thermosynechococcus_sp. did encompass a near-complete 16S rRNA gene, 99.6% similar to the *Thermosynechococcus elongatus* BP-1 homolog. This indicated that the nonappearance of *Thermosynechococcus* in 16S rRNA gene sequence based analyses was a bioinformatic glitch.

Discrepancies are known to arise in the microbial diversities revealed by analyzing the sample sample via shotgun metagenome sequencing and PCR-amplified 16S rRNA gene sequencing. A number of studies have found 16S amplicon sequencing to underrepresent taxa at different hierarchical levels [74–77], whereas the opposite outcome has also been reported by quite a few studies [78,79]. The present data demonstrated that the annotation of metagenomic reads corresponding to 16S rRNA genes reliably delineated most part of the diversity under investigation, at both the phylum and genus levels. Although this approach did not fare well in revealing the scarcely represented taxa, the diversity revealed by this approach was fairly

reproducible in the corresponding data obtained via taxonomic classification of CDSs annotated within the assembled metagenome, and the OTUs clustered from PCR-amplified 16S rRNA gene sequences. The latter two approaches, in turn, appeared to be more adept to unearthing the less-abundant taxa, at the phylum as well as genus level. With all the three approaches having their own strengths and limitations, a simultaneous use of all of them seems to be the best practice in obtaining a comprehensive picture of microbial diversity in a given environmental sample.

## Lotus Pond's vent-water microbiome remains stable over time

Over the last ten years, the Lotus Pond has been explored for its vent-water bacterial (but not archaeal) diversity, via 16S amplicon sequencing alone, on five previous time-points [10,17]. A comparison of the present 16S amplicon-based OTU data with those from the past showed that all the 16 bacterial phyla detected in the current investigation (S5 Table in S1 File), except Rhodothermota and Synergistetes, had been identified in previous exploration(s). The five previous explorations too had collectively discovered 16 bacterial phyla, of which only Ignavibacteriae and Nitrospirae went undetected in the present 16S amplicon-based survey. Furthermore, 11 bacterial phyla, namely Actinobacteria, Aquificae, Bacteroidetes, Chloroflexi, Cyanobacteria, Deinococcus-Thermus, Firmicutes, Proteobacteria, Synergistetes, Thermodesulfobacteria and Thermotogae were consistently detected in the present annotation of CDSs, 16S rRNA-encoding metagenomic reads, and OTUs clustered from 16S amplicon sequences; all of these, except Synergistetes, were detectable in one or more previous studies of 16S amplicon sequencing and analysis.

At the genus level, 121 entities were detected in the current PCR-based investigation (S5 Table in S1 File), out of which 59 were identified in one or more previous explorations, which in turn had collectively discovered 156 genera. Seven out of the 97 genera that were identified previously, but went undetected in the present PCR-based analysis (these were *Acidovorax*, Armatimonadetes_gp5, *Buttiauxella*, Cyanobacteria GpIV, *Klebsiella*, *Ornithinimicrobium* and *Sphingomonas*), got identified in the present annotation of metagenomic reads corresponding to 16S rRNA genes (S4 Table in S1 File). In this way, a total of 66 bacterial genera (Table 2) were found to be present consistently in Lotus Pond's vent-water over a period of approximately ten years. Notably, six out of these 66 genera, namely *Fervidobacterium*, *Halomonas*, *Paracoccus*, *Sulfurihydrogenibium*, *Tepidimonas* and *Thermus*, were unequivocally represented in the population genome bins (MAGs) obtained from the current metagenomic data, while two more, namely *Hydrogenobacter* and *Thiofaba*, were represented by very closely related entities, if not direct member strains of the genera. In this way, there were only two such bacterial genera—namely *Thermosynechococcus* and *Vibrio*—that figured among the MAGs obtained in the current study but were not represented in any of the previous OTU data.

## Lotus Pond is not reserved for thermophiles, albeit Aquificales predominate numerically and nutritionally

Literature survey for the phenotypic characteristics of the cultured members of the 92 bacterial genera identified via taxonomic classification of 16S rRNA encoding metagenomic reads showed that the constituent strains of only 29 genera had reports of growth in the laboratory above 45˚C (see S4 Table in S1 File and references therein). A similar exercise for the 121 genera detected via OTU-level clustering of PCR amplified 16S rRNA gene sequences showed that member strains of only 30 genera had reports of *in vitro* growth above 45˚C (S5 Table in S1 File). Among the 66 bacterial genera regularly present in Lotus Pond vent-water, only 15 were found to have strains reported for laboratory growth above 45˚C: these were

**Table 2. Bacterial genera that were detected in the present exploration of the Lotus Pond vent-water as well as in one or more previous studies of the same habitat using 16S rRNA gene sequence analysis.**

| Phylum to which the genera belonged | Names of the genera[1,2] | Maximum temperature for Laboratory-growth[3] (°C) | Sampling date of the previous study in which the genus was also identified | |
|---|---|---|---|---|
| | | | 23 July, 2013 [17] | 20 October, 2014 [10] |
| Actinobacteria | *Brevibacterium* | 42 | + | + |
| | *Corynebacterium* | 42 | + | + |
| | *Mycobacterium* | 43 | - | + |
| | *Nocardioides* | 37 | - | + |
| | *Ornithinimicrobium* | 50 | + | - |
| | *Rothia* | 38 | + | - |
| Armatimonadetes | Gp5 | - | - | + |
| | Gp7 | - | - | + |
| Aquificae | *Hydrogenobacter* | 85 | + | + |
| | *Sulfurihydrogenibium* | 80 | + | + |
| Bacteroidetes | *Chryseobacterium* | 37 | + | + |
| | *Cloacibacterium* | 40 | + | + |
| | *Pedobacter* | 35 | - | + |
| | *Porphyromonas* | 37 | + | - |
| | *Prevotella* | 45 | + | - |
| Chloroflexi | *Chloroflexus* | 59 | + | + |
| Cyanobacteria | GpXIII | - | - | + |
| | GpIV | - | - | + |
| Deinococcus-Thermus | *Thermus* | 80 | + | + |
| | *Truepera* | 50 | + | + |
| Dictyoglomi | *Dictyoglomus* | 80 | - | + |
| Firmicutes | *Anoxybacillus* | 70 | + | + |
| | *Bacillus* | 70 | + | + |
| | *Chryseomicrobium* | 45 | + | - |
| | *Enterococcus* | 45 | + | - |
| | *Exiguobacterium* | 49 | + | + |
| | *Gemella* | 37 | - | + |
| | *Lactococcus* | 40 | + | - |
| | *Lysinibacillus* | 45 | - | + |
| | *Planococcus* | 42 | + | + |
| | *Sporolactobacillaceae_incertae_sedis* | - | - | + |
| | *Staphylococcus* | 40 | + | + |
| | *Streptococcus* | 45 | + | + |
| Alphaproteobacteria | *Azospirillum* | 37 | + | + |
| | *Bradyrhizobium* | 37 | + | + |
| | *Brevundimonas* | 42 | + | + |
| | *Paracoccus* | 45 | + | + |
| | *Sphingobium* | 37 | - | + |
| | *Sphingomonas* | 45 | + | + |
| | *Sulfitobacter* | 40 | + | + |

*(Continued)*

**Table 2.** (Continued)

| Phylum to which the genera belonged | Names of the genera[1,2] | Maximum temperature for Laboratory-growth[3] (°C) | Sampling date of the previous study in which the genus was also identified | |
|---|---|---|---|---|
| | | | 23 July, 2013 [17] | 20 October, 2014 [10] |
| Betaproteobacteria | *Achromobacter* | 42 | + | + |
| | *Acidovorax* | 37 | + | + |
| | *Burkholderia* | 37 | + | + |
| | *Comamonas* | 44 | - | + |
| | *Delftia* | 40 | - | + |
| | *Herbaspirillum* | 45 | + | - |
| | *Pelomonas* | 37 | + | + |
| | *Ralstonia* | 41 | + | + |
| | *Tepidimonas* | 60 | + | + |
| Gammaproteobacteria | *Aeromonas* | 41 | + | + |
| | *Acinetobacter* | 37 | + | + |
| | *Buttiauxella* | 42 | - | + |
| | *Enhydrobacter* | 41 | + | + |
| | *Escherichia/Shigella* | 37 | - | + |
| | *Halomonas* | 50 | + | + |
| | *Klebsiella* | 37 | - | + |
| | *Marinobacter* | 50 | + | + |
| | *Pseudomonas* | 42 | - | + |
| | *Psychrobacter* | 38 | + | + |
| | *Rheinheimera* | 35 | - | + |
| | *Serratia* | 35 | + | - |
| | *Shewanella* | 30 | + | - |
| | *Stenotrophomonas* | 42 | + | + |
| | *Thiofaba* | 51 | + | + |
| | *Thiovirga* | 34 | + | - |
| Thermotogae | *Fervidobacterium* | 90 | + | + |

[1] Genera detected in the present PCR-amplified 16S rRNA gene sequence analysis are indicated by bold fonts.

[2] Names of the genera that were detected in the present annotation of metagenomic reads corresponding to 16S rRNA genes are underscored.

[3] Full references concerning the maximum temperatures for laboratory-growth of all the above genera have been cited in relation to S4 and S5 Tables in S1 File, within the Supplementary Information file.

*Fervidobacterium, Hydrogenobacter, Sulfurihydrogenibium, Thermus, Dictyoglomus, Anoxybacillus, Bacillus, Tepidimonas, Chloroflexus, Thiofaba, Truepera, Halomonas, Marinobacter, Ornithinimicrobium* and *Exiguobacterium* (Table 2). Corroborative to this high diversity of bacterial mesophiles, 41.2% of all taxonomically classifiable CDSs identified in the assembled metagenome were ascribable to Proteobacteria, whereas only 24.5% were affiliated to Aquificae (S3 Table in S1 File; Fig 3A). Interestingly, at least 70 CDSs for different cold shock proteins (most of them were DNA-binding) could be detected in the Lotus Pond metagenome, out of which, again, 22 were ascribable to Aquificae, two each to Deinococcus-Thermus and Thermotogae, and one to Thermodesulfobacteria; the rest were affiliated to Proteobacteria (38), Firmicutes (5), Actinobacteria (1), and Synergistetes (1). These genomic features corroborate that the Lotus Pond vent-water is neither a "Thermophiles only" habitat nor a closed, biophysically firewalled, ecosystem. Instead, it hosts physiologically diverse microorganisms, including

mesophiles and psychrophiles, which come into the system from the adjoining territories of the cold and arid Changthang Plateau. Furthermore, it highlights the potential fluxes of the thermophilic bacteria out of the high temperature ecosystem, into the cold desert biome, over time and space. All these suppositions are in agreement with the biogeodynamic model of the Puga hydrothermal system [10], as well as the "hot spring–river" ecological continuum that embodies the topography of the Lotus Pond site (Fig 1).

Although phylogenetic relatives of mesophilic bacteria overwhelmed the taxonomic, as well as the overall genetic, diversity of the Lotus Pond vent-water microbiome, thermophilic Aquificales dominated the habitat in terms of relative abundance. Only two MAGs affiliated to Aquificaceae/*Hydrogenobacter* (maximum temperature for laboratory-growth of Aquificaceae is 95˚C [80]), and *Sulfurihydrogenibium* (maximum temperature for laboratory-growth: 80˚C [81]), collectively made up 59.4% of the whole metagenome (Table 1 and S6 Table in S1 File). *Hydrogenobacter* and *Sulfurihydrogenibium* also accounted for ~80% of all 16S rRNA-encoding reads identified within the quality-filtered metagenomic sequence dataset (S4 Table in S1 File).

Among the other taxa identified as MAGs, *Fervidobacterium* (maximum temperature for laboratory-growth: 90˚C [82]), *Thermus* (maximum temperature for laboratory-growth: 80˚C [83]), the family Thermodesulfobacteriaceae of the phylum Thermodesulfobacteria (maximum temperature for laboratory-growth: 100˚C [84]), and the order Desulfurococcales within the class Thermoprotei of the phylum Crenarchaeota (maximum temperature for laboratory-growth: 110˚C [85]) have strains growing at temperatures above or within the ten-year range (78–85˚C) recorded for Lotus Pond's vent-water [10,16,17]. These MAGs together accounted for ~3% of the metagenome.

Cyanobacteria have been found in 73–85˚C springs [86,87], but *Thermosynechococcus* are restricted to ≤70˚C waters [88,89] while its cultured strains are known to grow only at ≤60˚C [90,91]. *Halomonas* [92], *Paracoccus* [28,93], *Tepidimonas* [94], Chromatiales [95] and *Vibrio* [96] have temperature maxima for laboratory-growth at 50˚C, 45˚C (or 50˚C under special conditions), 60˚C, 51˚C, and 50˚C respectively. Thus, eight out of the 15 Lotus Pond MAGs, collectively accounting for 5.6% of the metagenome, were affiliated to genera encompassing strains that grow only at temperatures below the range known for Lotus Pond's vent-water. That said, *Halomonas*, *Paracoccus*, *Tepidimonas*, *Thiofaba* and *Vibrio* have been isolated from hydrothermal vents or vent-adjacent environments, and found to exhibit varying degrees of thermal endurance [30,94,95,97–99].

The Aquificae not only predominated Lotus Pond's vent-water microbiome in terms of relative abundance, but also acted as the key primary producers of the ecosystem, accounting for most of the genes present in the metagenome for the most prevalent pathways of chemolithotrophy (sulfur oxidation) and autotrophy (carbon fixation), namely the Sox mechanism, and the rTCA cycle and the 3-hydroxypropionate bi-cycle, respectively. While the Sox multienzyme complex system is thought to have originated in ancient thermophilic bacteria ancestral to *Aquificae* and *Epsilonproteobacteria* [100], the putatively primordial rTCA cycle has supposedly originated in sulfide-rich hydrothermal vent environments of the early Earth [101].

Although the constituent protein subunits of Sox have coevolved parallel to one another as orthologous lineages in many extant sulfur chemolithotrophs, the system has also spread across the taxonomic spectrum of *Bacteria* through extensive paralogy of the components [100,102]. On the other hand, in the extant biota, the rTCA cycle is known to operate in members of Aquificales, and Chlorobi, Deltaproteobacteria, Epsilonproteobacteria, Nitrospirae, and Crenarchaeota [103–106]. Notably, the present exploration revealed the presence of rTCA cycle homologs not only in MAGs affiliated to Aquificaceae, but also those ascribed to

unclassified Bacteria, Desulfurococcales, *Fervidobacterium*, *Paracoccus*, *Tepidimonas*, *Thermus*, and *Vibrio*. The occurrence of rTCA cycle genes in the unexpected residents of this hydrothermal habitat (i.e. the phylogenetic relatives of known mesophiles) shed new light on the ecological amplitude, selection, and evolution (including horizontal spread) of this ancient autotrophic pathway in the context of thermal stress.

The preponderance of Sox and rTCA in Lotus Pond conforms not only to the pathways' ecology and evolution, but also the habitat's physical chemistry, spatial abundance and isotopic ratios of redox sulfur species, and the copious presence of $HCO_3^-$ [10,16]. In contrast, the 3-hydroxypropionate bi-cycle, discovered in the green non-sulfur bacterium called *Chloroflexus aurantiacus* [107,108], has long been considered as an exclusive feature of Chloroflexi. Recent studies, however, have shown that the pathway, or its analogous variants, could exist beyond this single phylum [109,110]. The present investigation of the Lotus Pond metagenome showed that genes for a few key enzymes of the 3-hydroxypropionate bi-cycle were present in the MAGs affiliated to Aquificae. However, since the pathway was not found in its complete form in these MAGs, it would be premature to vouch for the actual operation of 3-hydroxy-propionate bi-cycle in the members of Aquificae native to Lotus Pond.

Furthermore, in this context, it is noteworthy that the presence of a substantial cache of genes for the reductive pentose phosphate cycle in Lotus Pond's assembled metagenome as well as the MAGs belonging to *Tepidimonas*, *Thermosynechococcus* and unclassified Chromatiales is idiosyncratic to the biophysics of the ecosystem. This necessitates a renewed attention to the feasibility of this pathway at temperatures above the upper limit (70–75˚C) known thus far for its functionality [65]. Additionally, several homologs of (i) the phospho-enolpyruvate carboxylase gene *ppc* (K01595), (ii) the malate dehydrogenase *mdh* (K00024), (iii) the oxaloacetate decarboxylating and $NADP^+$ dependent malate dehydrogenase gene *maeB* (K00029), and (iv) the pyruvate, orthophosphate dikinase *ppdK* (K01006) were found to be parts of the Lotus Pond vent-water metagenome, as well as a few MAGs such as those belonging to *Paracoccus* and *Tepidimonas*. The protein products of these genes are known to govern the carbon fixation pathway called crassulacean acid metabolism (CAM) in plants adapted to aridity and high temperature stress [111]. To avoid gaseous exchange during the hot day time, CAM plants store $CO_2$ in the form of malic acid at night, transport the malate to chloroplasts during the day, re-convert it $CO_2$ and concentrate the same in the proximity of ribulose bisphosphate carboxylase molecules to carry out efficient carbon fixation [111]. A new line of pure culture based research of physiology and biochemistry needs to be initiated to explore the plausible roles of analogous mechanisms in bacteria adapted to thermal stress.

Lotus Pond's vent-water metagenome contained several loci for the biosynthesis of secondary metabolites including antibiotics, a large proportion of which was ascribable to Aquificae. Members of this phylum native to Lotus Pond were also endowed with a rich repertoire of genes for antibiotic resistance and multi-drug efflux pumping. Archaea as a whole, Deinococcus-Thermus, Thermodesulfobacteria, and Thermotogae accounted for a small percentage and diversity of the antibiotic biosynthesis and resistance related genes detected; mesophilic taxa led by Alphaproteobacteria and Gammaproteobacteria, however, possessed these genes in large numbers and varieties (S8 Table in S1 File). While the potential antibiosis machinery of the Lotus Pond Aquificae might help them overcome plausible niche overlap with their ecological equivalents (especially the thermophilic and hyperthermophilic archaea), such contrivances can also exacerbate the bioenergetic costs of thermal endurance for the phylogenetic relatives of mesophilic bacteria that infiltrate the Lotus Pond hydrothermal system under the mediation of the local geodynamic forces [10].

## Genomic resources that can help mesophiles survive in the Lotus Pond habitat

Most of the genetic diversity present in Lotus Pond's vent-water microbiome was contributed by phylogenetic relatives of mesophilic bacteria, even though they were not numerally predominant (as subsequently evident from the direct annotation of metagenomic reads corresponding to rRNA gene sequences) and were also plausibly not alive or metabolically active in large numbers. That said, Lotus Pond's assembled metagenome encompassed several such genes that could be potentially vital for thermal adaptation, especially in the context of the vent-water's geochemistry and the spring's topography (Supplementary Results). Also, a good majority of these CDSs were ascribable to phylogenetic relatives of mesophilic bacteria, and their most conspicuous examples included copious genes encoding heat shock proteins, molecular chaperones and chaperonin complexes that render the quality control of DNA and proteins amid biophysical stress. The metagenome contained several genes encoding proteins that control/modulate/inhibit DNA gyrase activity. In true thermophiles, reverse gyrases (RGs) are indispensible as they render positive supercoils in the DNA to counter-balance the negative supercoils promoted by gyrases, and *in situ* heat [112]. RGs also help avert incorrect aggregation of denatured DNA, and facilitate correct annealing [113]. Expectedly, Lotus Pond MAGs affiliated to Aquificaceae, Desulfurococcales, *Fervidobacterium* and *Sulfurihydrogenibium* encoded RGs, but those belonging to phylogenetic relatives of mesophilic or moderately thermophilic bacteria did not. Nevertheless, MAGs affiliated to *Halomonas*, *Paracoccus*, *Tepidimonas*, and *Vibrio* did encode gyrase inhibitors/modulators such as YacG/PmbA/TldD which are known to nullify gyrase activity [114,115]; notably, several homologs of these genes ascribable to other mesophilic taxa were also present in the assembled metagenome. Future studies of metatranscriptomics and metaproteomics are needed alongside multi-omic investigations of pure culture isolates to ascertain whether and how the above specialties help hot spring mesophiles endure high temperatures over time and space.

The metagenome encoded different toxin-antitoxin (TA) systems, including the Type II, Type IV, RelE/RelB, PrlF/YhaV, YdaS/YdaT, HicAB, and MraZ/MraW systems, which act as regulatory modules for growth and survival of bacterial cells under diverse stress conditions [116]. Other ecologically advantageous genes having large numbers of homologs in the assembled metagenome encoded the following: (a) universal stress proteins, homologs of which are also possessed by bacterial mesophiles occupying ecological niches adjacent to deep sea hydrothermal vents [98]; (b) methionine sulfoxide reductases (Msr), which repair oxidatively damaged/inactivated proteins by converting their methionine sulfoxides to methionines; (c) enzymes known to protect cells from the toxic effects of peroxides, which in turn are often abundant in hot spring waters [117]; (d) diverse proteins adapted structurally and/or functionally to high concentrations of heavy metals, which is a common feature of hydrothermal discharges [118–121] including those of Lotus Pond and other hot springs of Puga Valley [5,10,14–16]; (e) different Mnt proteins that render cellular transport and homeostasis of manganese, a key cofactor of different enzymes protecting against oxidative damage; (f) fatty acid desaturases that may help increase the rigidity of cell membranes (via unsaturation of fatty acids).

Numerous genes were also there in the Lotus Pond metagenome for flagellar structure/function, chemotaxis, cell adhesion/aggregation, biofilm formation, and quorum sensing. All these attributes have the potentials to assist Lotus Pond's native microorganisms respond to *in situ* environmental conditions and position themselves dynamically at their biophysically/biochemically best-suited niches along the hydrothermal (Fig 1A) and biogeochemical [10] gradients occurring in the vent to river (Rulang) trajectory.

## Why are virulence genes abundant in Lotus Pond?

The Lotus Pond metagenome encompassed several CDSs for the formation of biofilms, adhesion of cells, biosynthesis of hemagglutinin and hemolysin, and stationary phase metabolisms including persister cell formation (S2 Table in S1 File). While biofilm formation, cellular adhesion, and haemolysin synthesis may all have key roles in osmoregulation and extreme-temperature response [122,123], formation of persister cells can help bacterial mesophiles attain ultra-slow metabolic states, which have recently been reported to be central to the endurance of high *in situ* temperatures [30]. Furthermore, the metagenome encompassed more than a thousand genes for constructing Type-I through Type-IX SSs, excepting the Type-V and Type-VII variants. While all these molecular machineries secreting biomacromolecules to the cells' exterior have clearly documented roles in bacterial survival within eukaryotic hosts (and consequent virulence), their potentially multifaceted ecological roles are also not unappreciated [98]. For instance, macromolecules moved across the cell membrane by Type-II SS often serve as local and long-range effectors/promoters of nutrient acquisition and biofilm formation under adverse environmental conditions [124,125]. Type-III SS, besides translocating effector bacterial proteins into host cells, plays important roles in stress response including those involving high salts and peroxides in the chemical milieu of the bacterium [126]. Systems level network analyses have also revealed interrelations between Type-III and Type-IV SSs on one hand and stress management on the other [127]. Furthermore, Type-IV SS homologs are also sometimes involved in inter-bacterial killing, which can not only eliminate competition but also turn out to be beneficial for the killers in nutrient deprived habitats [128,129]. Type-VI SS homologs, besides being central to survival within eukaryotic hosts, regulate quorum sensing with wide-ranging implications for environmental adaptation [130,131]. Type-VIII SS is implicated in adhesion, aggregation, and biofilm formation [132], while the Type-IX SS, reported thus far from *Bacteroidetes* alone, confers gliding motility to the environmental strains of the phylum [133]. Thus, in the context of the mesophiles present in the Lotus Pond vent-water, both these SSs might turn out to be instrumental in placing the native bacteria to biophysically favorable sites within the hydrothermal gradients (Fig 1A).

## Summing up

High temperature habitats despite being the extant analogs of primordial ecosystems on Earth are situated at the functional extremities of the biosphere. They are characterized by such biophysical conditions which constrain habitability for most of the extant microbial taxa via entropic disordering of biomacromolecules that in turn imposes unsustainably high energetic costs on the maintenance of the metabolic machinery. In such a scenario, only those few specialized groups of thermophilic microorganisms, which can afford the high energy-cost of maintenance and avoid cell system failure due to high heat, manage to colonize hydrothermal ecosystems and prevail *in situ*. That said, hot springs are not totally out of bound territories for free-living environmental microorganisms that do not have any phylogenetic (taxonomic) relative known for *in vitro* growth at >45˚C. For instance, Trans-Himalayan hot spring waters poor in calcium and silicon, but rich in boron, chlorine, sodium, and various sulfur species, are known based on PCR-amplified 16S rRNA gene sequence data to harbor high diversities of bacterial mesophiles that are stochastically introduced to the system by local geodynamic forces. The present study for the first time undertook deep shotgun metagenome sequencing and analysis for such a high-biodiversity Trans-Himalayan hot spring called Lotus Pond to comprehensively reveal the microbiome architecture of its boiling vent-water discharge.

The gross microbial cell density in Lotus Pond's vent-water was found to be $8.5 \times 10^4$ mL$^{-1}$. Approximately 63.5% of this microbial population (i.e. $5.4 \times 10^4$ cells mL$^{-1}$ vent-water) was found

to be metabolically active as the cells could take in FDA and hydrolyze it to fluorescein that eventually was detected via fluorescence microscopy. Assembly, annotation, and population-binning of >15-GB metagenomic sequence elucidated the relative abundance, metabolisms, and adaptations of all the microbial groups present in the 85°C fluid vented by the sulfur-borax spring called Lotus Pond, and in doing so illuminated the numeral predominance of Aquificae (particularly, *Hydrogenobacter* and *Sulfurihydrogenibium*), even as Proteobacteria contributed most of the genetic diversity of the microbiome; prevalence and diversity of Deinococcus-Thermus, Thermotogae, Thermodesulfobacteria, Crenarchaeota and Euryarchaeota were far less. Perpetual presence of bacterial mesophiles in Lotus Pond's vent-water was evident from the detection of 51 such genera that had been discovered in previous 16S amplicon-based investigations of the habitat, but have no member with laboratory growth at >45°C. Within the Lotus Pond vent-water microbiome maximum genetic resource was devoted to Translation, ribosomal structure and biogenesis, followed by Energy production and conversion. Lotus Pond's metagenome was replete with genes crucial for high temperature adaptation, especially in the context of Lotus Pond's geochemistry and topography. By sequence homology, a majority of these genes were attributable to phylogenetic relatives of mesophilic bacteria. Copious genes within the metagenome were also directly or indirectly related to virulence functions and antibiotics synthesis/resistance. Since a good proportion of these genes were attributable to Aquificae, they could well help the group outcompete other thermophiles present *in situ*, and contain the proliferation of the mesophilic intruders of the ecosystem. That said, in the absence of culture-based microbiological data and gene expression analyses, the actual activity of the Lotus Pond microflora in this regard remains purely speculative and subject to future testimony. In the same vein, there is still no conclusive evidence to prove that the mesophiles present in the Trans-Himalayan sulfur-borax hot springs contribute to the live microflora *in situ*. In that context, however, it is noteworthy that the holistic elucidation of microbiome structure and functions for the boiling vent-waters of Trans-Himalayan sulfur-borax hot springs is an ongoing process. We had previously outlined the alpha diversity of the system via analysis of amplified 16S rRNA gene sequences [10,17,18], then we isolated a few strains that were phylogenetically related to mesophilic bacteria [17,30]; here we explored the microbiome via deep shotgun metagenomics, and in the near future we shall communicate the findings of metatranscriptome analysis.

## Supporting information

**S1 File. An MS Word file named S1 File, which contains a comprehensive table of contents enumerating all the supporting information accompanying this paper, one Supplementary Method, three Supplementary Results, four Supplementary Tables, Supplementary References, and one Supplementary Figure.**
(DOC)

**S1 Dataset. An MS Excel file named S1 Dataset, which contains four Supplementary Tables that are all more than one page in breadth.**
(XLSX)

## Acknowledgments

We thank Professor Shubhra Ghosh Dastidar and Sri Sanjib Kumar Gupta of Bose Institute, Kolkata, for helping with computational resources.

## Author Contributions

**Conceptualization:** Wriddhiman Ghosh.

**Data curation:** Nibendu Mondal, Wriddhiman Ghosh.

**Formal analysis:** Nibendu Mondal, Subhajit Dutta, Sumit Chatterjee, Jagannath Sarkar, Mahamadul Mondal, Ranadhir Chakraborty, Wriddhiman Ghosh.

**Funding acquisition:** Wriddhiman Ghosh.

**Investigation:** Nibendu Mondal, Subhajit Dutta, Sumit Chatterjee, Jagannath Sarkar, Mahamadul Mondal, Wriddhiman Ghosh.

**Methodology:** Nibendu Mondal, Subhajit Dutta, Sumit Chatterjee, Jagannath Sarkar, Chayan Roy, Wriddhiman Ghosh.

**Project administration:** Wriddhiman Ghosh.

**Resources:** Wriddhiman Ghosh.

**Software:** Nibendu Mondal, Subhajit Dutta, Jagannath Sarkar, Chayan Roy.

**Supervision:** Wriddhiman Ghosh.

**Validation:** Nibendu Mondal, Subhajit Dutta, Sumit Chatterjee, Jagannath Sarkar, Mahamadul Mondal, Chayan Roy, Ranadhir Chakraborty, Wriddhiman Ghosh.

**Visualization:** Nibendu Mondal, Subhajit Dutta, Sumit Chatterjee, Jagannath Sarkar, Chayan Roy, Ranadhir Chakraborty, Wriddhiman Ghosh.

**Writing – original draft:** Wriddhiman Ghosh.

**Writing – review & editing:** Nibendu Mondal, Ranadhir Chakraborty, Wriddhiman Ghosh.

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
