## [Decision Letter · Decision Letter 0]

27 Dec 2023

PONE-D-23-36526Antibiosis helps Aquificae surmount competition by archaeal thermophiles, and crowding by bacterial mesophiles, to dominate the boiling vent-water of a Trans-Himalayan sulfur-borax springPLOS ONE

Dear Dr. Ghosh,

Thank you for submitting your manuscript to PLOS ONE. After careful consideration, we feel that it has merit but does not fully meet PLOS ONE’s publication criteria as it currently stands. Therefore, we invite you to submit a revised version of the manuscript that addresses the points raised during the review process.

We look forward to receiving your revised manuscript.

Kind regards,

Durgesh Kumar Jaiswal, Ph.D.

Academic Editor

PLOS ONE

Journal Requirements:

1. When submitting your revision, we need you to address these additional requirements. Please ensure that your manuscript meets PLOS ONE's style requirements, including those for file naming. The PLOS ONE style templates can be found at https://journals.plos.org/plosone/s/file?id=wjVg/PLOSOne_formatting_sample_main_body.pdf and https://journals.plos.org/plosone/s/file?id=ba62/PLOSOne_formatting_sample_title_authors_affiliations.pdf 2. We suggest you thoroughly copyedit your manuscript for language usage, spelling, and grammar. If you do not know anyone who can help you do this, you may wish to consider employing a professional scientific editing service.  Whilst you may use any professional scientific editing service of your choice, PLOS has partnered with both American Journal Experts (AJE) and Editage to provide discounted services to PLOS authors. Both organizations have experience helping authors meet PLOS guidelines and can provide language editing, translation, manuscript formatting, and figure formatting to ensure your manuscript meets our submission guidelines. To take advantage of our partnership with AJE, visit the AJE website (http://aje.com/go/plos) for a 15% discount off AJE services. To take advantage of our partnership with Editage, visit the Editage website (www.editage.com) and enter referral code PLOSEDIT for a 15% discount off Editage services. If the PLOS editorial team finds any language issues in text that either AJE or Editage has edited, the service provider will re-edit the text for free. Upon resubmission, please provide the following:  The name of the colleague or the details of the professional service that edited your manuscript A copy of your manuscript showing your changes by either highlighting them or using track changes (uploaded as a *supporting information* file)  A clean copy of the edited manuscript (uploaded as the new *manuscript* file) 3. In your Methods section, please provide additional information regarding the permits you obtained for the work. Please ensure you have included the full name of the authority that approved the field site access and, if no permits were required, a brief statement explaining why. 4. Thank you for stating the following in the Acknowledgments Section of your manuscript: This research was financed by Bose Institute (via intramural faculty grants) as well as the Science and Engineering Research Board (SERB), Government of India (GoI) (SERB grant number EMR/2016/002703). N.M. received fellowships from SERB and Bose Institute. S.D. and J.S. obtained their fellowships from Council of Scientific and Industrial Research, GoI. S.C. and M.M. received a fellowship from the Department of Biotechnology, GoI. We thank Professor Shubhra Ghosh Dastidar and Sri Sanjib Kumar Gupta for helping with computational resources. We note that you have provided funding information that is not currently declared in your Funding Statement. However, funding information should not appear in the Acknowledgments section or other areas of your manuscript. We will only publish funding information present in the Funding Statement section of the online submission form. Please remove any funding-related text from the manuscript and let us know how you would like to update your Funding Statement. Currently, your Funding Statement reads as follows: WG received funds for this research from Bose Institute (http://www.jcbose.ac.in/home) via intramural faculty grants as well as the Science and Engineering Research Board (SERB, https://serb.gov.in/), Government of India (GoI) (SERB grant number EMR/2016/002703). N.M. received fellowships from SERB and Bose Institute. S.D. and J.S. obtained their fellowships from Council of Scientific and Industrial Research, GoI (https://www.csir.res.in/). S.C. and M.M. received a fellowship from the Department of Biotechnology, GoI (https://dbtindia.gov.in/). The funders played no role in the study design, data collection and analysis, decision to publish, or preparation of the manuscript. Please include your amended statements within your cover letter; we will change the online submission form on your behalf.

Additional Editor Comments:

Dear Authors,

After carefully evaluated the manuscript, we find that the study is very interesting however, before considering the manuscript for publication is required the major revision. So we have requested the authors to follow the reviewers comments and improvised the manuscript according it.

Thanks

Reviewers' comments:

Reviewer's Responses to Questions

**Comments to the Author**

1. Is the manuscript technically sound, and do the data support the conclusions?

Reviewer #1: Partly

Reviewer #2: No

Reviewer #3: No

Reviewer #4: Partly

2. Has the statistical analysis been performed appropriately and rigorously? 

Reviewer #1: N/A

Reviewer #2: N/A

Reviewer #3: I Don't Know

Reviewer #4: N/A

3. Have the authors made all data underlying the findings in their manuscript fully available?

Reviewer #1: Yes

Reviewer #2: Yes

Reviewer #3: Yes

Reviewer #4: Yes

4. Is the manuscript presented in an intelligible fashion and written in standard English?

Reviewer #1: No

Reviewer #2: No

Reviewer #3: Yes

Reviewer #4: Yes

5. Review Comments to the Author

Reviewer #1: I wish to express my gratitude to the author for their diligent efforts in conducting the study titled "Antibiosis helps Aquificae surmount competition by archaeal thermophiles, and crowding by bacterial mesophiles, to dominate the boiling vent-water of a Trans-Himalayan sulfur-borax spring". It contributes to enhance our understanding on functional profile of microbiome from geothermal region. However, there is a significant need for improvement in the method section, particularly considering the nature of the sequencing platform. All claims in the study rely solely on results from gene calling, assembly, taxonomic assignment, and functional annotation. Substantial enhancements in the methodological approach would strengthen the study's validity and reliability.

I have outlined certain aspects that require attention from the authors.

I encountered difficulty in understanding the English used, particularly in the abstract and introduction. Sentences are excessively long, featuring unnecessary words and a lack of coherence between phrases separated by commas. Please aim for clarity and conciseness.

The author utilized the eggNOG database for functional analysis, last updated in 2019. However, it's worth noting that the COG database (https://www.ncbi.nlm.nih.gov/research/cog) is the most recent, updated in March 2022. The COG database provides dedicated resources for both bacteria and archaea. I strongly recommend that the author considers using the COG database for more up-to-date and comprehensive functional profiling.

The author might want to consider including Quality Check (QC) results of the FASTQ files in the supplementary section, both before and after read filtration. This additional information can offer insights into the data quality and the impact of the filtration process.

Consistency should maintain in providing parameter information for the tools used in the analysis. For example parameter information for Prodigal v2.6.3 is missing. If default parameters were used, please mention this in the documentation.

The study references the searching of reads against the rrnDB database, a crucial step for taxonomic delineation. However, there is limited information on read mapping and extraction. It would be beneficial if the author could provide a detailed workflow, including steps, parameters, and intermediate files used for the mapping and extraction of reads. Additionally, if any other tools were employed in this process, their inclusion as an workflow would be appreciated.

Given that the Shotgun sequence is not the most suitable option for taxonomic profiling with publicly available 16S databases, I suggest that the authors utilize a dedicated pipeline for shotgun metagenome data, such as MEGAN Community Edition.

The study employed the UBCG Pipeline v3.0 for core gene phylogeny, which has limitations in identifying less core gene due to its reliance on a universal core genome database.Consequently, it overlooked several actual core genes in the user dataset. I recommend utilizing the best-bidirectional hits (BBH) method for core gene identification, as it is sure to offer a more comprehensive core gene set from the given genomic dataset.

MAGs with accession numbers JARJND000000000 and JARJNC000000000 exhibit a completeness of less than 50% in Table-1, indicating significant incompleteness and poor quality. It is advisable to exclude these MAGs from the analysis for more robust and reliable results.

Reviewer #2: This manuscript presents data from a unique ecosystem that is understudied and worthy of better characterisation. For that reason, I am recommending a major revision before this manuscript can be published.

I had a hard time reading this manuscript due both to the way it is written and the choice of data analysis and visualisation.

The manuscript is excessively long, which compromises readability. Throughout the paper the tendency for long sentences, that usually focus on different ideas, is maintained. In many sentences the choice of words is unusual, to say the least.

As an example of what I am saying the sentence in the Introduction:

"Idiosyncratic to the above community structure axiomatic for hydrothermal ecosystems" - what is an axiomatic ecosystem, and what idiosyncrasy are you referring to?

Please consider getting someone to help revising the English. As it is, it reads as if a machine has tried to come up with dictionary synonyms for words that are more commonly used, even in scientific writing.

There is a lack of detail in the Material and Methods section regarding data analysis and visualisation. You detail how you get MAGs and OTUs and then no more information is provided. After you get the MAGs and the 16S reads how do you analyse them? For example, compare the legend in Fig 3 and Fig 6, the latter explains the analysis that was done.

Also asking as this is not the most common way of presenting these data. It is also very uninformative, as none of your Figures help to visualise the conclusions you draw from your data. Figure 5 is particularly unhelpful, and if I understood correctly the only thing that is related to your work is the legend?

Your data analysis and visualisation should show and support the main points you make in your conclusions. As an example, in your title you use the word Antibiosis, but nowhere in the paper do you show that is possible. The fact that you find a potential gene does not mean that it is being expressed. You have no data from RNA, therefore any activity must be considered as potential. Either you do that, or you rephrase the manuscript to reflect the data analysis that you are presenting.

Please note that these comments are meant to be read as examples of issues with the whole manuscript, which should be thoroughly revised to improve readability, reduce length and provide data visualisation and analysis that is in line with the most common methods.

Reviewer #3: The paper by Mondal et al. is devoted to the microbiome of the hot spring (namely, geyser), called Lotus Pond, located in the Puga Valley on the Changthang plateau. The authors applied the metagenomics analysis, including MAGs binning and their functional analysis, 16S rRNA amplicon sequencing and enumeration of microbial cells, filtered out from the hot spring water, with different fluorescent dyes. The main hypotheses of this work are:

- that genomes of Aquificota possess many genes of antibiotic biosynthesis and even virulence functions, as well as antibiotic resistance genes, that helps them dominating in the hot spring, overlapping the other thermophiles, especially Archaea;

- and bacterial mesophiles are not random components of this hot spring community and their genomes are replete with genes crucial for hydrothermal and other stress adaptation;

However, the data, obtained by in silico analysis of MAGs, does not support these conclusions.

Please, find all comments and observations in the attached document

Reviewer #4: Interesting and very well written manuscript. I have only one reservation - you analyzed just the metagenome, so, the conclusions of the article and partially even the title of the article are not based on experimental data. The presence of a large number of "resistance" genes is not an evidence that these genes are actually expressed. For many bacterial taxa, it is well documented that these genes or gene clusters are not actually expressed.

Without additional transcriptomic data, in my opinion, the title of the article is inappropriate

6. PLOS authors have the option to publish the peer review history of their article (what does this mean?). If published, this will include your full peer review and any attached files.

Reviewer #1: **Yes: **Sushanta Deb

Reviewer #2: No

Reviewer #3: No

Reviewer #4: **Yes: **Peter Pristas

---

## [Author Response · Author response to Decision Letter 0]

11 Mar 2024

Response to the comments of Reviewer 1

I wish to express my gratitude to the author for their diligent efforts in conducting the study titled "Antibiosis helps Aquificae surmount competition by archaeal thermophiles, and crowding by bacterial mesophiles, to dominate the boiling vent-water of a Trans-Himalayan sulfur-borax spring". It contributes to enhance our understanding on functional profile of microbiome from geothermal region. However, there is a significant need for improvement in the method section, particularly considering the nature of the sequencing platform. All claims in the study rely solely on results from gene calling, assembly, taxonomic assignment, and functional annotation. Substantial enhancements in the methodological approach would strengthen the study's validity and reliability.

RESPONSE: We thank the Reviewer for appreciating our detailed exploration of the microbiome architecture of the geobiologically unique Trans-Himalayan sulfur-borax spring system. 

We agree with the suggestions pertaining to the strengthening of the methods section, so have now dealt with these concerns primarily by bringing to the main text all those procedural details that were previously narrated in the Supplementary Methods. Thus, in the revised Article File, 

we have now elaborated the previous Methods section called “Metagenomics” under three new sections titled as

- Metagenome extraction and sequencing

- Assembly and annotation of the metagenomic sequence

- Direct annotation of metagenomic reads corresponding to rRNA gene sequences

besides significantly substantiating the Methods sections

- Construction and taxonomic characterization of metagenome-assembled genomes (MAGs)

- PCR-amplified 16S rRNA gene sequence analysis.

I have outlined certain aspects that require attention from the authors.

I encountered difficulty in understanding the English used, particularly in the abstract and introduction. Sentences are excessively long, featuring unnecessary words and a lack of coherence between phrases separated by commas. Please aim for clarity and conciseness.

RESPONSE: We agree with the Reviewer’s concerns regarding the apparently difficult English used in certain sections of the manuscript. We have now tried our best to make the sentences simple and short, yet comprehending some of them may still require additional attention. The geochemical distinctiveness of the hot spring explored and the uniqueness of the microbiome thriving therein are both quite complex. Accordingly, the intricate issues pertaining to the geomicrobiological peculiarities of Trans-Himalayan sulfur-borax springs, compared with other well studied hydrothermal ecosystems, need to be put in their proper perspectives without over-simplification.

The author utilized the eggNOG database for functional analysis, last updated in 2019. However, it's worth noting that the COG database (https://www.ncbi.nlm.nih.gov/research/cog) is the most recent, updated in March 2022. The COG database provides dedicated resources for both bacteria and archaea. I strongly recommend that the author considers using the COG database for more up-to-date and comprehensive functional profiling.

RESPONSE: We agree, so have now removed all COG-related information from the eggNOG-derived annotation of CDSs shown in Table S2 (previously numbered as Table S1), and provided fresh data obtained by searching the Prodigal-derived putatively translated gene catalog against the COG Little Endian version of the Conserved Domain Database of NCBI located at https://ftp.ncbi.nih.gov/pub/mmdb/cdd/little_endian/Cog_LE.tar.gz, using COGclassifier v1.0.5 (please see lines 241-245, 498-517 of the Track Changes file). 

The author might want to consider including Quality Check (QC) results of the FASTQ files in the supplementary section, both before and after read filtration. This additional information can offer insights into the data quality and the impact of the filtration process.

RESPONSE: We agree, so have now included Quality Check (QC) results for the raw read set as well as the one obtained after quality filtering (please see the new Table S1 of the Revised Manuscript).

Consistency should maintain in providing parameter information for the tools used in the analysis. For example parameter information for Prodigal v2.6.3 is missing. If default parameters were used, please mention this in the documentation.

RESPONSE: We agree and have now included the parameter information for all the bioinformatic software programs that have been used in this study, including Prodigal v2.6.3 (please see lines 197-321 of the Track Changes file).

The study references the searching of reads against the rrnDB database, a crucial step for taxonomic delineation. However, there is limited information on read mapping and extraction. It would be beneficial if the author could provide a detailed workflow, including steps, parameters, and intermediate files used for the mapping and extraction of reads. Additionally, if any other tools were employed in this process, their inclusion as an workflow would be appreciated.

RESPONSE: We agree and have now added the warranted details of read mapping, extraction, and classification, to the new Methods section titled “Direct annotation of metagenomic reads corresponding to rRNA gene sequences” (please see lines 254-261 of the Track Changes file).

Given that the Shotgun sequence is not the most suitable option for taxonomic profiling with publicly available 16S databases, I suggest that the authors utilize a dedicated pipeline for shotgun metagenome data, such as MEGAN Community Edition.

RESPONSE: We used the MEGAN6 Community Edition to taxonomically classify the metagenomic reads that were detected by searching the raw dataset against the SILVA_SSURef_138.1_NR99 database with the help of Minimap2-2.26 (Li, 2018). Curiously, this led to the detection of no read ascribed to Crenarchaeota / Thermoproteota, Thermotogae / Thermotogota, Cyanobacteria, Chloroflexi / Chloroflexota, Actinobacteria / Actinomycetota, Synergistetes / Synergistota, even though 396, 196, 131, 56, 21, and 2 reads were detected for these taxa respectively, in our existing RDP-based analyses. Betaproteobacteria and Alphaproteobacteria, for which 1345 and 595 reads were identified in the RDP-based annotation, also went undetected in the MEGAN-based analysis. In fact, no phylum other than Aquificota, Proteobacteria (only the Gamma class), Firmicutes, Deinococcota, Bacteroidota were detected in the MEGAN6-based classification.

Given that the metagenome under consideration is that of a boiling vent-water community, in situ presence of phyla such as Thermoproteota (this included genera such as Aeropyrum, Pyrobaculum and Stetteria) and Thermotogota, as detected in the existing RDP-based classification, is obviously true; accordingly, a bioinformatic tool which fails to identify these native taxa cannot constitute a technique of choice in the present case.

In the revised manuscript, therefore, we retain the contextually more acceptable RDP-based classification of raw metagenomic reads corresponding to 16S rRNA genes.

The study employed the UBCG Pipeline v3.0 for core gene phylogeny, which has limitations in identifying less core gene due to its reliance on a universal core genome database. Consequently, it overlooked several actual core genes in the user dataset. I recommend utilizing the best-bidirectional hits (BBH) method for core gene identification, as it is sure to offer a more comprehensive core gene set from the given genomic dataset.

RESPONSE: We agree that the UBCG Pipeline identifies relatively less number of core genes due to its reliance on the universal core gene database. However, since the MAG dataset involved in the current analysis includes entities diverged at the level of the phylum, we have no other choice but to use the universal core gene database (hence UBCG) to accommodate all the MAGs in one phylogenetic tree. Furthermore, when we used Roary - The pan genome pipeline v3.23.0, as well as GET_HOMOLOGUES v3.6.2, to identify core genes using the best-bidirectional hits (BBH) method, we were unable to identify any gene with a conservation level of >90% sequence similarity across the MAGs. 

MAGs with accession numbers JARJND000000000 and JARJNC000000000 exhibit a completeness of less than 50% in Table-1, indicating significant incompleteness and poor quality. It is advisable to exclude these MAGs from the analysis for more robust and reliable results.

RESPONSE: It’s true that the completeness levels of the MAGs having accession numbers JARJND000000000 and JARJNC000000000 are 30% and 34% respectively. However, it is noteworthy that the two MAGs have contamination levels of only 0.3% and 1.1% respectively, so they cannot be regarded as being poor in quality.

In the current analysis, 19, 14, and 30 MAGs were initially generated using Metabat2, MaxBin2, and CONCOCT, respectively. Comprehensive refinement and optimization of the 63 population genome bins taken together, using DASTool, finally qualified 15 MAGs as the best representatives of the metagenome, notwithstanding the low completeness level of a few shortlisted MAG. This result needs to be presented upfront in its entirety without any modification/subtraction of convenience. This is important for the sake of the comprehensiveness as well as the integrity of the data. 

Moreover, in the context of the microbiome, it is most important to note that the two MAGs JARJND000000000 and JARJNC000000000 belonged to the family Thermodesulfobacteriaceae and the genus Paracoccus respectively. Thermodesulfobacteria is the most predominant taxon of true thermophiles within the Lotus Pond ecosystem after Aquificae, while Paracoccus strains isolated during previous expeditions of Lotus Pond have been studied in-depth for their thermal endurance (Mondal et al. 2022: Microbiology Spectrum 10, e01606-22).

All the above issues strongly support the essentiality of a robust documentation of the two MAGs in the present manuscript, even if their completeness is currently low (notably, generation of more metagenomic sequence data from this sample in the future can well increase the completeness of the two MAGs).

Response to the comments of Reviewer 2

This manuscript presents data from a unique ecosystem that is understudied and worthy of better characterisation. For that reason, I am recommending a major revision before this manuscript can be published.

I had a hard time reading this manuscript due both to the way it is written and the choice of data analysis and visualisation.

RESPONSE: We agree, and believe that the Reviewer appreciated our detailed exploration, as well as the underlying phenomenology, of the microbiome architecture of the geobiologically unique Trans-Himalayan hot spring system. As suggested, we have now overhauled the manuscript by focusing on the main findings and making the overall text (especially the language) more lucid.

The manuscript is excessively long, which compromises readability. Throughout the paper the tendency for long sentences, that usually focus on different ideas, is maintained. In many sentences the choice of words is unusual, to say the least.

As an example of what I am saying the sentence in the Introduction:

"Idiosyncratic to the above community structure axiomatic for hydrothermal ecosystems" - what is an axiomatic ecosystem, and what idiosyncrasy are you referring to?

RESPONSE: We have now tried our best to abridge the text, remove the asides, and make the sentences simple and short. Yet following some of the sentences may still require additional attention. The geochemical distinctiveness of the hot spring explored and the uniqueness of the microbiome thriving therein are both quite complex. Accordingly, the issues pertaining to their peculiarities (idiosyncrasies), compared with other well studied hydrothermal systems and their microbiota, need to be put in their proper perspectives without over-simplification.

As for the sentence in question, we have now assorted the information into two separate sentences, and added precise terminologies (such as “low-biodiversity community structure” instead of “above community structure”, and “extra-ordinarily diversified vent-water microflora” instead of “extra-ordinarily high habitability”) to bring more clarity to the message conveyed (please see lines 93-98 of the Track Changes file). 

In relation to the Reviewer’s queries “what is an axiomatic ecosystem", and “what idiosyncrasy had been referred to”, a careful reading of the previous sentence

“Idiosyncratic to the above community structure axiomatic for hydrothermal ecosystems, a geochemically unusual category of Trans-Himalayan hot springs …... have been reported to feature extra-ordinarily high habitability”

points out that 

- no ecosystem is being called axiomatic here but the typical community structure described in the previous paragraph is being termed as axiomatic for hydrothermal ecosystems in general (please note the preposition “for” after the word “axiomatic”);

- and, the extra-ordinarily high habitability of Trans-Himalayan hot springs is being referred to as idiosyncratic to the aforesaid community structure that in turn is axiomatic for hydrothermal ecosystems (in this case the preposition “to” after the word “idiosyncratic” is to be noted).

Please consider getting someone to help revising the English. As it is, it reads as if a machine has tried to come up with dictionary synonyms for words that are more commonly used, even in scientific writing.

RESPONSE: We have now considerably shortened the text by deleting the issues of tangential importance (please see lines 65-80, 646-653, 686-692, 746-798, 838-876, 886-913, 1182-1237 of the Track Changes file). The new text focuses on the main findings and their implications. We have also made the sentences simpler by splitting them wherever possible. That said, some sentences may still found complicated and warrant additional attention as the issues dealt there are complex and cannot be oversimplified.

There is a lack of detail in the Material and Methods section regarding data analysis and visualisation. You detail how you get MAGs and OTUs and then no more information is provided. After you get the MAGs and the 16S reads how do you analyse them? For example, compare the legend in Fig 3 and Fig 6, the latter explains the analysis that was done.

RESPONSE: In the previous manuscript, all the details of the bioinformatic procedure were given in the Supplementary Methods. We have now brought all those critical information to the main text to add to the reader’s satisfaction as well as the study's reliability (please see lines 167-321 of the Track Changes file).

Thus, in the revised Article File, 

we have now elaborated the previous Methods section called “Metagenomics” under three new sections titled as

- Metagenome extraction and sequencing

- Assembly and annotation of the metagenomic sequence

- Direct annotation of metagenomic reads corresponding to rRNA gene sequences,

besides significantly substantiating the Methods sections

- Construction and taxonomic characterization of metagenome-assembled genomes (MAGs)

- PCR-amplified 16S rRNA gene sequence analysis (for the upgraded methods pertaining to MAGs and OTUs please see lines 263-321 of the Track Changes file).

Also asking as this is not the most common way of presenting these data. It is also very uninformative, as none of your Figures help to visualise the conclusions you draw from your data. Figure 5 is particularly unhelpful, and if I understood correctly the only thing that is related to your work is the legend?

RESPONSE: In the revised manuscript, 

Figure 1 depicts the topography of the study site. Figure 2 shows representative picture of the fluorescence microscopic fields based on which microbial cell density was calculated in the vent-water sample of Lotus Pond. Figure 3 illustrate the overall microbiome composition by providing an overview of the taxonomic distribution of (A) the CDSs identified in the assembled metagenome of Lotus Pond, and (B) the metagenomic reads which corresponded to 16S rRNA genes. Figure 4 presents the phylogen

---

## [Decision Letter · Decision Letter 1]

3 Sep 2024

Aquificae overcomes competition by archaeal thermophiles, and crowding by bacterial mesophiles, to dominate the boiling vent-water of a Trans-Himalayan sulfur-borax spring

PONE-D-23-36526R1

Dear Dr. Ghosh,

We’re pleased to inform you that your manuscript has been judged scientifically suitable for publication and will be formally accepted for publication once it meets all outstanding technical requirements.

Kind regards,

Marcos Pileggi, Ph.D

Academic Editor

PLOS ONE

Additional Editor Comments (optional):

Reviewers' comments:

Reviewer's Responses to Questions

**Comments to the Author**

1. If the authors have adequately addressed your comments raised in a previous round of review and you feel that this manuscript is now acceptable for publication, you may indicate that here to bypass the “Comments to the Author” section, enter your conflict of interest statement in the “Confidential to Editor” section, and submit your "Accept" recommendation.

Reviewer #1: All comments have been addressed

2. Is the manuscript technically sound, and do the data support the conclusions?

Reviewer #1: Yes

3. Has the statistical analysis been performed appropriately and rigorously? 

Reviewer #1: Yes

4. Have the authors made all data underlying the findings in their manuscript fully available?

Reviewer #1: Yes

5. Is the manuscript presented in an intelligible fashion and written in standard English?

Reviewer #1: Yes

6. Review Comments to the Author

Reviewer #1: (No Response)

7. PLOS authors have the option to publish the peer review history of their article (what does this mean?). If published, this will include your full peer review and any attached files.

Reviewer #1: **Yes: **Sushanta Deb

---

## [Editor Report · Acceptance letter]

16 Oct 2024

PONE-D-23-36526R1 

PLOS ONE

Dear Dr. Ghosh, 

I'm pleased to inform you that your manuscript has been deemed suitable for publication in PLOS ONE. Congratulations! Your manuscript is now being handed over to our production team.

Kind regards, 

on behalf of

Dr. Marcos Pileggi 

Academic Editor

PLOS ONE